# On the Entropy Calibration of Language Models

**Steven Cao**
Stanford University
shcao@stanford.edu

**Gregory Valiant**
Stanford University
valiant@stanford.edu

**Percy Liang**
Stanford University
pliang@cs.stanford.edu

## Abstract

We study the problem of entropy calibration, which asks whether a language model's entropy over generations matches its log loss on human text. Past work found that models are miscalibrated, with entropy per step increasing as generations grow longer, due to error accumulation. To calibrate the model and improve text quality, it has become standard practice to truncate the distribution, but this approach reduces output diversity, which we would like to avoid. Therefore, in this paper, we ask: does miscalibration improve automatically with scale, and if not, is it theoretically possible to calibrate without tradeoffs? To build intuition, we first study a simplified theoretical setting to characterize the scaling behavior of miscalibration with respect to dataset size. We find that the rate of scaling depends on the power law exponent of the data distribution — in particular, for a power law exponent close to 1, the scaling exponent is close to 0, meaning that miscalibration improves very slowly with scale. Next, we measure miscalibration empirically in language models ranging from 0.5B to 70B parameters. We find that the observed scaling behavior is similar to what is predicted theoretically: our fitted scaling exponents for text are close to 0, meaning that larger models accumulate error at a similar rate as smaller ones. This scaling (or, lack thereof) provides one explanation for why we sample from larger models with similar amounts of truncation as smaller models, even though the larger models are of higher quality. However, truncation is not a satisfying solution because it comes at the cost of increased log loss. In theory, is it even possible to reduce entropy while preserving log loss? We prove that it is possible, if we assume access to a black box which can fit models to predict the future entropy of text.

## 1 Introduction

We study entropy calibration, which asks whether a language model's entropy over generations matches its log loss on human text. This definition is a natural notion of calibration for generative tasks, and is more challenging than calibration for classification tasks because the output space is exponentially large. Braverman et al. (2020) were the first to study entropy calibration in language models, and they found that models are miscalibrated: entropy per step for model generations increases with the length of the document, in contrast with log loss on human text which is roughly flat over the length of the document (Genzel & Charniak, 2002; Verma et al., 2023). Entropy calibration can be thought of as a quantitative formalization of the well-known "error accumulation" or "teacher forcing" problem: entropy rises when the model generates erroneous tokens which are fed back into the context, derailing the generation (Williams & Zipser, 1989; Ranzato et al., 2016; Basu et al., 2021; Hewitt et al., 2022, also see Appendix E). Therefore, we study entropy calibration to gain fundamental insights into improving generation quality.

To correct error accumulation and calibrate the model, it has become standard practice to truncate the next token distribution (Fan et al., 2018; Holtzman et al., 2020; Hewitt et al., 2022), suppressing low

---

https://github.com/stevenxcao/entropy-calibration

39th Conference on Neural Information Processing Systems (NeurIPS 2025).

probability tokens to improve quality at the cost of diversity (Hashimoto et al., 2019; Zhang et al., 2021). This solution is not satisfying: diversity is especially important for difficult tasks where we must aggregate multiple answers (Wang et al., 2024; Brown et al., 2024), as well as for synthetic data generation, which has seen a resurgence of interest as the community has begun worrying about running out of internet data (Wang et al., 2023; Gunasekar et al., 2023; Maini et al., 2024). Therefore, it is natural to ask: do we expect miscalibration to improve with scale? If not, is it at least theoretically possible to calibrate without sacrificing diversity?

To build intuition, we first study a simplified theoretical setting, where instability comes from the fact that the model might generate a token that it saw only a few times during training. This unfamiliar token then derails subsequent steps when it is fed back into the context autoregressively. Drawing on classic results, we calculate a scaling exponent capturing how quickly the probability of generating a rare token decreases with the number of training examples (Good, 1953; Karlin, 1967). We find that this exponent depends on how heavy-tailed the data distribution is: in particular, for power law exponents close to 1, as is typical for human text (Zipf, 1936, 1949), the scaling exponent is close to 0. Therefore, this setup predicts that stability in generation improves very slowly with scale.

Next, we measure miscalibration empirically in large language models with up to 70B parameters, on three datasets. We find that the observed scaling behavior is similar to what is predicted by the simplified setting: fitting scaling exponents relating calibration error to model size, we find that the exponent for the two text datasets is around $-0.05$, meaning that larger models are similarly miscalibrated as smaller ones. On the other hand, for the code dataset, the scaling exponent is around $-0.3$, meaning that miscalibration improves moderately with scale. We measure the power law exponent to be around 1 for the two text datasets, and 1.5 for the code dataset. Therefore, these findings are consistent with the theory: the code dataset has more quickly decaying tails, so the scaling should indeed be faster. However, further work on more datasets is needed to more strongly establish this relationship between the power law and scaling exponents.

If even large models suffer from error accumulation, why are reasoning and instruction-tuned models able to produce long, coherent outputs? We find that much like distribution truncation, instruction tuning reduces entropy at the cost of increased log loss, with the largest models now having entropy too low. This tradeoff relates to past work which found that alignment degrades model capabilities, a phenomenon known as the alignment tax (Ouyang et al., 2022; Bai et al., 2022; Lin et al., 2024).

Given that all known mitigations increase the model's log loss, is it even possible in theory to calibrate without this tradeoff? Drawing on ideas from reinforcement learning theory, we prove that it is possible, if we assume access to a black box which can fit models on the future entropy of text prefixes and attain low test error. Specifically, we describe a polynomial-time calibration procedure that adjusts each candidate token's probability based on the expected entropy of its continuations. While the resulting procedure is impractical to implement, we prove that it calibrates while preserving log loss, suggesting that generation stability and diversity might be possible to attain simultaneously.

## 2 Preliminaries

We first review key definitions and properties for entropy calibration, introduced in Braverman et al. (2020). Our setup is as follows: we are given prompts $X \in \mathcal{X}$ drawn from some prompt distribution $X \sim q$, and responses $Y \in \mathcal{Y}$ drawn from the true conditional distribution $Y \sim p_X^*$. For example, $X$ might contain a description of a coding task, while $Y$ contains a solution to the task. We then train a language model $\hat{p} : \mathcal{X} \to \Delta^{\mathcal{Y}}$ to fit the true conditional distribution $p^*$. We say that $\hat{p}$ is *entropy calibrated* if its entropy over generations is equal to its log loss:

$$H(\hat{p}) = \mathcal{L}(p^* \| \hat{p}), \tag{1}$$

where the total/per-step entropy and total/per-step log loss are given by

$$H(\hat{p}) = \mathbb{E}_{X \sim q} \mathbb{E}_{\hat{Y} \sim \hat{p}_X}[-\log \hat{p}_X(\hat{Y})], \qquad H_t(\hat{p}) = \mathbb{E}_{X \sim q} \mathbb{E}_{\hat{Y} \sim \hat{p}_X}[-\log \hat{p}_X(\hat{Y}_t \mid \hat{Y}_{<t})] \tag{2}$$

$$\mathcal{L}(p^* \| \hat{p}) = \mathbb{E}_{X \sim q} \mathbb{E}_{Y \sim p_X^*}[-\log \hat{p}_X(Y)], \quad \mathcal{L}_t(p^* \| \hat{p}) = \mathbb{E}_{X \sim q} \mathbb{E}_{Y \sim p_X^*}[-\log \hat{p}_X(\hat{Y}_t \mid \hat{Y}_{<t})]. \tag{3}$$

To build intuition for this definition, entropy can be thought of as a measure of the model's uncertainty, which should be calibrated to match the actual loss it incurs on real data. This definition mirrors that of binary calibration, and we derive this connection more formally in Appendix B. Qualitatively, if a model is underconfident, then its generations have too much entropy and appear incoherent;

if it is overconfident, then its generations have too little entropy and appear repetitive (Braverman et al., 2020; Basu et al., 2021); see Appendix E for a replication of this finding and Appendix D for examples. Entropy calibration is then the problem of adjusting the entropy to be just right. Empirically, Braverman et al. (2020) found that neural autoregressive language models have entropy too high: entropy per step matches the log loss at earlier steps but increases as the generation grows.

Why does entropy per step grow with the length of the generation? The main problem, as has been observed in empirical work, is that autoregressive language models accumulate error during generation. At training time, models are given input from the true distribution and asked to produce only a single additional token. In contrast, models must generate multiple tokens at deployment time, which they do by producing one token at a time and taking their own production as subsequent input. Therefore, even models with very low single-step error can degrade over multiple steps as they take their own slightly erroneous outputs as input and accumulate errors (see, e.g., Ranzato et al. (2016), Welleck et al. (2020), Holtzman et al. (2020) for error accumulation in language modeling; and Daumé et al. (2009), Ross & Bagnell (2010), Ross et al. (2011) for imitation learning). This intuition is formalized in the context of entropy calibration in Corollary 4.2 of Braverman et al. (2020), which states that for a model with $\varepsilon$ KL divergence to the true distribution, the entropy at step $t$ can deviate as much as $\varepsilon + \sqrt{\varepsilon t}$ from the log loss, growing with $t$.

How does one calibrate the entropy? Unlike binary and multiclass calibration, entropy calibration is challenging because the models have an exponentially large output space. In practice, practitioners use a number of distribution truncation methods, each of which uses a different heuristic to suppress low probability tokens in each generation step. Some standard methods include temperature reduction, top-k sampling (Fan et al., 2018), top-p sampling (Holtzman et al., 2020), and min-p sampling (Hewitt et al., 2022). These methods improve text quality at the cost of diversity (Hashimoto et al., 2019; Zhang et al., 2021; Pillutla et al., 2021; Welleck et al., 2024). Following Hashimoto et al. (2019), we define a model's diversity to be its log loss on reference documents. The intuition behind this definition is that log loss (also known as cross entropy or forward KL) is a coverage metric: to attain low log loss, the model must "cover" as much as the reference distribution as possible. Our goal, then, is to calibrate entropy to match log loss without also causing the log loss to increase.

In theory, Braverman et al. (2020) show that one can calibrate entropy while preserving log loss via globally normalized temperature scaling, where the adjusted model is given by $\hat{p}_\tau(y_1, ..., y_L) \propto \hat{p}(y_1, ..., y_L)^{1/\tau}$. Unfortunately, this adjustment is intractable to compute because it involves normalizing over the entire output space. It remains unclear, then, whether this goal is possible in polynomial time. Specifically, we wish to take in a model $\hat{p}$ and produce a calibrated model $\tilde{p}$ with at most $\varepsilon$ entropy calibration error per step, as well as log loss at most that of the original model $\hat{p}$:

$$\frac{1}{T} \left| \text{EntCE}(p^* \parallel \tilde{p}) \right| \leq \varepsilon, \tag{4}$$

$$\mathcal{L}(p^* \parallel \tilde{p}) \leq \mathcal{L}(p^* \parallel \hat{p}), \tag{5}$$

where the entropy calibration error is defined as the difference between the entropy and the log loss:

$$\left| \text{EntCE}(p^* \parallel \hat{p}) \right| = \left| H(\hat{p}) - \mathcal{L}(p^* \parallel \hat{p}) \right|. \tag{6}$$

## 3 Intuition: Singleton Mass in Power Law Data

Before putting in the work to develop better calibration algorithms, it is natural to first ask whether we expect miscalibration to automatically improve with scale, as we train larger models on more data. To gain intuition, we first explore this question in a simplified theoretical setting. We define the setup to capture the following hypothesis regarding error accumulation (see, e.g., Hewitt et al. (2022)): because the language distribution is heavy-tailed, the model must assign non-zero probability to a large number of rare tokens when fitting the data to achieve low log loss. However, if it happens to generate one such rare token, the model derails when that token is fed back into the context autoregressively, leading to a jump in entropy. Over many generation steps, then, the model will eventually derail. The degree of instability then depends on the probability of producing a rare token.

Accordingly, our setup is as follows: at training time, the model stores the counts for $m$ tokens drawn i.i.d. from an $\alpha$-power-law distribution $p$ over a vocabulary of size $v$, defined as $p_i \propto 1/i^\alpha$ for $i = 1, ..., v$. The model then generates a sequence token-by-token as follows: if all tokens in context were seen at least twice at training time, the model samples a random token seen during training.

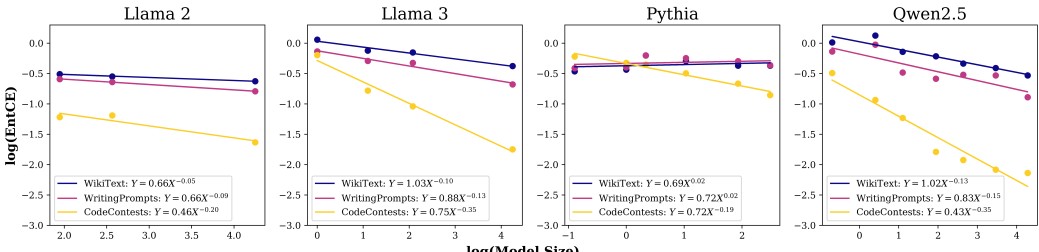

Figure 1: Log calibration error versus log model size for four model families and three datasets. We find that the scaling laws fit relatively well, suggesting that the relationship between calibration and scale is predictable. Furthermore, while there is variation between model families, the scaling exponents for each dataset are somewhat close to those predicted by theory (WikiText: $0.089$, WritingPrompts: $-0.10$, CodeContests: $-0.33$), with heavier-tailed datasets having slower scaling.

But if any token in context was seen only once, the model samples from a high entropy "derailed" distribution instead. This simple stylized setting captures our intuition about error accumulation and lets us study the effect of $\alpha$, representing the heavy-tailedness of the data distribution.

In this setting, the expected entropy per step grows with slope proportional to the probability of emitting a rare token (see Appendix B). Computing the rare token mass in power law data is a classic problem, and we can compute the asymptotic scaling exponent with respect to the number of training examples $m$ as follows (Good, 1953; Karlin, 1967):

**Proposition 3.1** (informal). *For $v$ infinite and $m$ large, the per-step probability of generating a singleton, in expectation over draws of the training set, is given by*

$$\mathbb{E}\frac{K_{m,1}}{m} = C_\alpha m^{1/\alpha - 1},$$

*where $C_\alpha$ is a constant depending only on $\alpha$, and $K_{m,1}$ is a random variable denoting the number of items seen exactly once in a set of $m$ samples.*

We provide the derivation in the appendix. The key takeaway from this proposition is that the derailing probability scales as $m^{1/\alpha - 1}$, which is very slow if the power law exponent $\alpha$ is close to 1, as is typical for text (Zipf, 1936, 1949). The reason for this slow scaling is that as $m$ increases, there are always more rare items to be sampled from the tail of the distribution. In practice, of course, we are not training unigram models, but the same intuition holds if we posit that semantic concepts in text are similarly heavy tailed: as larger models are trained on more data, there will be always new rare phenomena that they see during training only once. These phenomena are then memorized, and potentially generated at deployment, derailing the model.

While asymptotic analysis gives us a clean expression, we can also estimate the scaling exponent in simulation for finite values of $m$ and $v$. We find that the non-asymptotic simulated slopes are close to the asymptotic expression as long is $m$ is smaller than $v/3$ (see Appendix E). We also calculate the power law exponent for our three datasets, finding that it is around 1 for WikiText and WritingPrompts and 1.5 for CodeContests, which predicts slow scaling for the first two datasets and slow-to-moderate scaling for the third.

## 4 Experiments: Miscalibration in Large Language Models

Next, we measure miscalibration empirically in large language models. We study four model families (**Qwen2.5** (0.5B, 1.5B, 3B, 7B, 14B, 32B, 72B) (Qwen et al., 2025), **Llama 3** (1B, 3B, 8B, 70B) (Grattafiori et al., 2024), **Llama 2** (7B, 13B, 70B) (Touvron et al., 2023), and **Pythia** (410M, 1.4B, 2.8B, 6.9B, 12B) (Biderman et al., 2023)) applied to the three datasets listed below. In each setting, we use 5000 examples and limit samples to 1024 tokens; see Appendix C for more experimental details. We primarily study base models because we are interested in the problem of modeling human text; we study the effect of instruction tuning in Section 4.3.

(a) **WikiText-103** (Merity et al., 2017): given 128 tokens of context from a Wikipedia passage, the model is tasked with completing the passage.

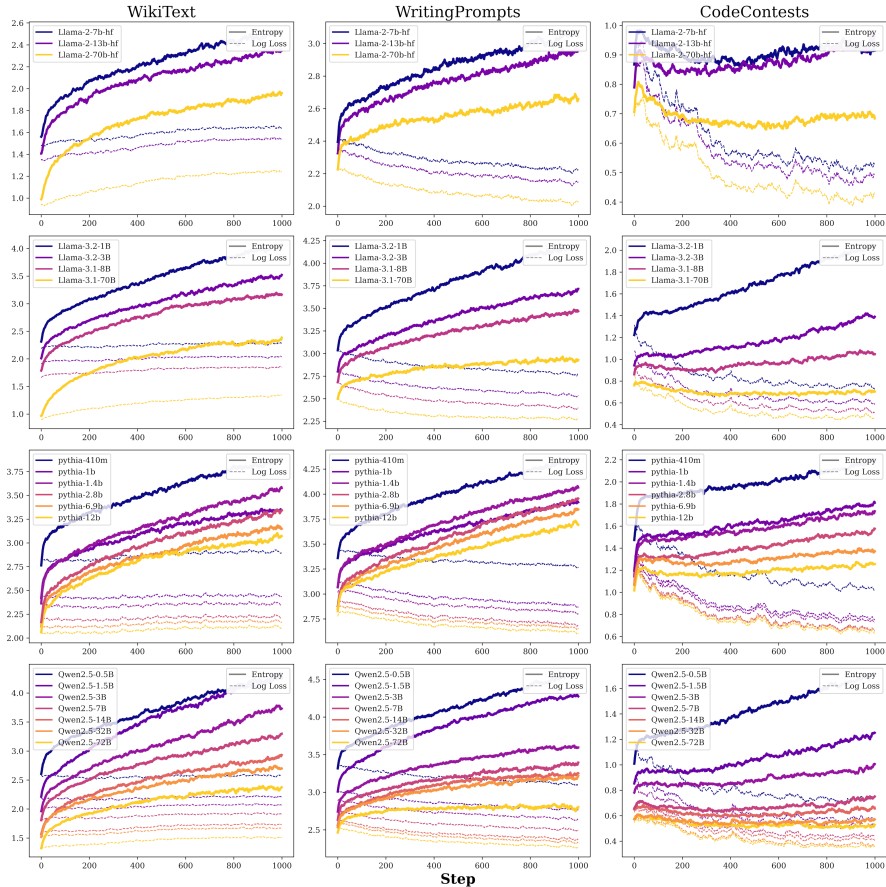

Figure 2: Entropy for each generation step (solid) and log loss for each token in the ground truth (dashed), for each dataset (columns) and each model family (rows), with models colored by size. Models have entropy much higher than their log loss, with the gap growing with the number of generation steps, a sign of error accumulation. For the text datasets, models of different sizes seem to be similarly miscalibrated, while for code the degree of miscalibration seems to improve with size.

(b) **WritingPrompts** (Fan et al., 2018): given a prompt from r/writingprompts along with 128 tokens of context from a human-written story, the model is tasked with completing the story.

(c) **CodeContests** (Li et al., 2022): given a coding problem from one of five websites and 128 tokens of context from a human-written solution, the model is tasked with completing the solution.

## 4.1 Miscalibration scaling in base models

Past work has found that many model capabilities improve predictably with model size, with task loss and model size following a linear relationship when plotted on a log scale (Kaplan et al., 2020; Hoffmann et al., 2022). We use a similar methodology to study the relationship between entropy miscalibration and model size. If model size and dataset size are scaled proportionally, Proposition 3.1 suggests a scaling law of $\log \text{EntCE} = (1/\alpha - 1) \log m + C$, where $\alpha$ is the power law exponent of the data distribution and $m$ is the parameter count. Does the actual data also follow a clean scaling law, and how close is the scaling exponent to that predicted by the simplified setting?

For each model-dataset combination, we compute the model's calibration error as the difference between its average entropy per generation step and its average log loss on ground truth data. We then plot log calibration error versus log model size, as shown in Figure 1.

First, we find that the linear fit is accurate, suggesting that the relationship between calibration and scale is predictable. Next, we find that the scaling exponents are dataset-dependent: for the older model families (Llama 2 and Pythia), the exponents are around $0.0$ for WikiText and WritingPrompts

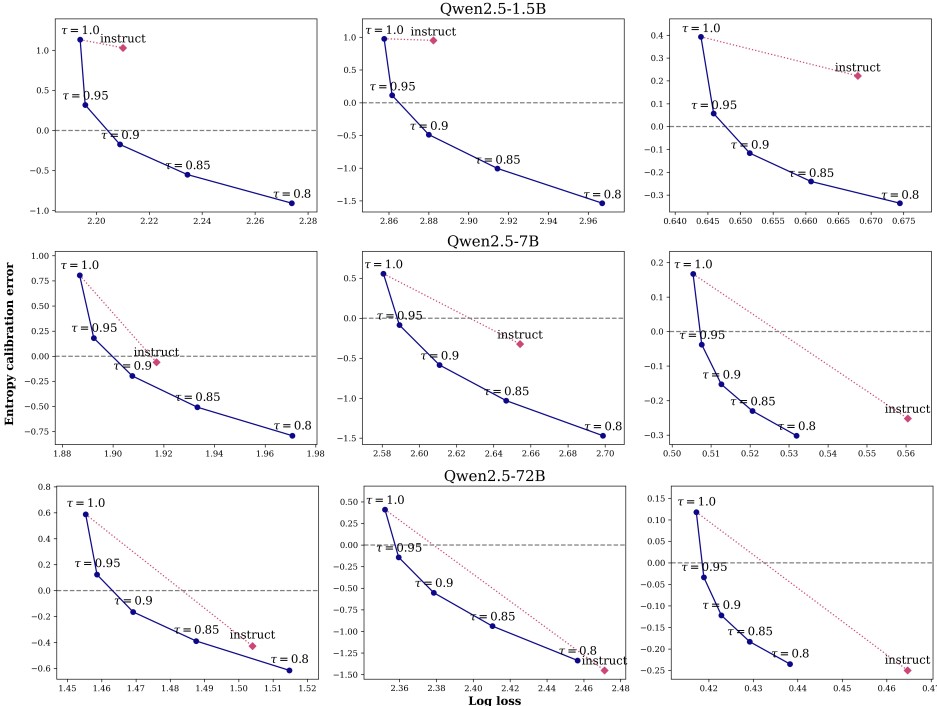

Figure 3: Entropy calibration error versus log loss for base Qwen2.5 (1.5B, 7B, 72B) compared to the instruction-tuned versions, along with various temperature settings (please see Appendix E for all model sizes). Positive calibration error means that the model's entropy is higher than its log loss, while negative means that its entropy is lower than its log loss. We find that each modification of the base model reduces entropy while increasing log loss, calibrating at the cost of diversity.

and $-0.2$ for CodeContests, while for the newer model families (Llama 3 and Qwen2.5), the exponents are around $-0.13$ for WikiText and WritingPrompts and $-0.35$ for CodeContests. Notably, these exponents are somewhat close to what is predicted theoretically (Figure 4): WikiText and WritingPrompts, with power law exponents of $0.918$ and $1.114$, are predicted to have slow scaling exponents of $0.089$ and $-0.10$, while CodeContests, with a power law exponent of $1.5$, is predicted to have a moderate scaling exponent of $-0.33$. However, future work on more datasets would be needed to more strongly establish the relationship between these exponents empirically.

We speculate that recent model families have better scaling due to changes in their pretraining data mixtures, and especially the addition of a midtraining step with higher quality and less diverse data. However, training details for three out of the four model families (all but Pythia) are not public, and future work with controlled data mixtures would be useful to disentangle the effects of model size, dataset size, and dataset composition.

Overall, these plots suggest that miscalibration in text generation improves very slowly with scale: a scaling exponent of $-0.10$ means that to reduce calibration error by a factor of 10, dataset size must increase by a factor of $10^{10}$.

## 4.2 Entropy over time

To gain a more fine-grained understanding of entropy blowup, we also produce entropy over time plots for each model and each dataset, as shown in Figure 2. Specifically, we plot each model's entropy at each generation step $t$, averaged over 5000 generated samples. We then compare this curve to the model's log loss on each token $t$ of a ground truth example, averaged over 5000 examples. Recall from Section 2 that theoretically, for an accurate model, entropy is initially close to log loss, but can deviate as much as $\sqrt{t}$ at the $t$-th step. A calibrated model which does not experience error accumulation should have entropy close to the log loss for all generation steps.

First, we find that for each model and dataset, the log loss is mostly constant or slightly decreasing over time. Past papers use a model's log loss to estimate the actual entropy of the underlying text, as the former is an upper bound for the latter that grows tighter if the model is more accurate. This part of the plot replicates past findings that the entropy per step of human text is stable over time, also known as the entropy rate constancy principle (Genzel & Charniak, 2002; Verma et al., 2023).

On the other hand, unlike human text, the entropy per step of model generations is not stable and instead increases over time. The lack of scaling shown quantitatively in the previous subsection is reflected visually in Figure 2, with larger models having entropy growing at similar rates as smaller models for WikiText and WritingPrompts (the left and middle columns). For CodeContests, the slopes decrease with model size, visually confirming that there is slow-to-moderate scaling.

## 4.3 Calibration-diversity tradeoffs

In this subsection, we seek to better understand how distribution truncation and post-training affect entropy. For each model-dataset combination (excluding Pythia, which has no instruction-tuned version), we compare the model with temperature $1.0$ to that with temperature $0.95$, $0.9$, $0.85$, or $0.8$, as well as to the instruction-tuned version of the model. We then plot entropy calibration error against log loss, where each setting of the model is one point on the plot, as shown in Figure 3.

First, we find that reducing temperature below 1 reduces entropy but increases log loss, replicating similar findings in past work (Hashimoto et al., 2019). Furthermore, the temperature attaining zero calibration error is similar across model sizes, which makes sense given that they are similarly miscalibrated. We find that instruction tuning also reduces entropy while increasing log loss, which is consistent with past work showing that instruction tuning harms diversity (Ghosh et al., 2024). Unlike temperature scaling, the magnitude of the effect varies across model sizes, with larger models experiencing both a larger reduction in entropy and larger increase in log loss. However, this pattern is not robust across model families (see Appendix E). Further work with more controlled instruction tuning would be necessary to explore this relationship further. These experiments reconcile our previous finding, that even large models accumulate errors, with the fact that in practice, one can use truncation or post-training to generate long, coherent pieces of text. The tradeoff is that each of these mitigations comes at the cost of diversity.

# 5 Theory: Future Entropy Scaling

If all known mitigations increase log loss, is it even possible in theory to calibrate without this tradeoff? In this section, we provide evidence that this tradeoff is not inevitable: given the assumption that there exists a procedure to fit regression models that generalize to i.i.d. test data, we show that there exists a tractable, albeit impractical, procedure that calibrates while preserving log loss.

## 5.1 Definitions

For a model $\hat{p}_X(Y_1, ..., Y_T)$ mapping a prompt $X$ to a distribution $\Delta^{\mathcal{Y}}$ over the output space $\mathcal{Y} = \mathcal{V}^T$, let the *future entropy* of the prefix $(X, Y_{\leq t})$ be given by

$$H_{\hat{p}_X}(Y_{>t} \mid Y_{\leq t}) = \sum_{Y_{>t}} -\hat{p}_X(Y_{>t} \mid Y_{\leq t}) \log \hat{p}_X(Y_{>t} \mid Y_{\leq t}). \tag{7}$$

Given a prefix $Y_{\leq t}$, this expression computes the model's entropy over the remaining generation $Y_{>t}$. We can then define the *future entropy adjusted* model, for parameters $\alpha = (\alpha_1, ..., \alpha_T) \in \mathbb{R}^T$, as

$$\hat{p}_{\alpha;X}^{(\text{ent})}(Y_t \mid Y_{<t}) \propto \exp\left\{(1 + \alpha_t) \log \hat{p}_X(Y_t \mid Y_{<t}) - \alpha_t H_{\hat{p}_{\alpha;X}^{(\text{ent})}}(Y_{>t} \mid Y_{\leq t})\right\}. \tag{8}$$

This expression adjusts each candidate token's probability based on what the future entropy would be if that token were chosen. The calibration procedure then involves fitting models to predict the future entropy of prefixes, and choosing the weights $\alpha_t$ to calibrate the model (Algorithm 1).

## 5.2 Assumptions

For a distribution $\hat{p}$ that can be tractably sampled from, we can take a Monte Carlo estimate to compute the future entropy, which concentrates because entropy is bounded (Algorithm 2). However, we cannot

**Algorithm 1** Future entropy scaling

**Inputs:** model $\hat{p}$, length $T$, vocab $\mathcal{V}$, future entropy fitting algorithm $\mathcal{A}$, future entropy dataset size $n$, sample size $m$, prompt distribution $q$, true conditional distribution $p^*$, optimization tolerance $\varepsilon$

1: Initialize $\alpha_1 = ... = \alpha_T = 0$, $\hat{f}_2 = ... = \hat{f}_{T+1} = 0$.

2: For $t = T, ..., 1$:

3:    Choose $\alpha_t$ to minimize expected log loss at step $t$, until the gradient is at most $\varepsilon$:

$$\alpha_t = \operatorname*{argmin}_{\alpha'_t} \mathcal{L}_t \left( p^* \parallel \hat{p}^{(\text{ent})}_{\alpha', \hat{f}} \right)$$

   where $\alpha' = (0, ..., 0, \alpha'_t, \alpha_{t+1}, ..., \alpha_T)$. ($\mathcal{L}_t$: Equation 3, $\hat{p}^{(\text{ent})}_{\alpha', \hat{f}}$: Equation 9).

4:    Fit the future entropy predictor $\hat{f}_t$ as follows:

5:        Sample prefixes $\left( X^{(i)}, Y^{(i)}_{<t-1} \right)_{i=1}^n \sim (q, p^*)$.

6:        For each token $v \in \mathcal{V}$, compute labels $(h^{(i,v)})_{i=1}^n$ by passing each prefix $\left( X^{(i)}, \left[ Y^{(i)}_{<t-1}, v \right] \right)$ into Algorithm 2, along with inputs $\hat{p}^{(\text{ent})}_{\alpha, \hat{f}}$, $T$, $m$.

7:        Fit one future entropy predictor for each token $v$, setting $\hat{f}_t(X, [Y_{<t-1}, v]) = \hat{f}_{t,v}(X, Y_{<t-1})$, where each $\hat{f}_{t,v}$ is the output $\mathcal{A} \left\{ \left( X^{(i)}, Y^{(i)}_{<t-1}, h^{(i,v)} \right)_{i=1}^n \right\}$.

8: Return $(\alpha_1, ..., \alpha_T), (\hat{f}_2, ..., \hat{f}_{T+1})$.

---

**Algorithm 2** Future entropy estimation via sampling

**Inputs:** model $\hat{p}$, length $T$, prefix $(X, Y_{\leq t})$, samples $m$

1: Sample $m$ trajectories from the model: $\left( \hat{Y}^{(i)}_{t+1}, ..., \hat{Y}^{(i)}_T \right)_{i=1}^m \overset{\text{i.i.d.}}{\sim} \hat{p}_X(\hat{Y}_{>t} \mid Y_{\leq t})$.

2: Return $\hat{H} = \frac{1}{m} \sum_{i=1}^m \sum_{s=t+1}^T - \log \hat{p}_X(\hat{Y}^{(i)}_s \mid \hat{Y}^{(i)}_{<s})$.

---

assume that $\hat{p}^{(\text{ent})}_{\alpha; X}$ can be tractably sampled from, so we cannot compute its future entropy naively. Instead, we will use our assumed model fitting procedure to iteratively replace each intractable future entropy term $H_{\hat{p}^{(\text{ent})}_{\alpha; X}}(Y_{>t} \mid Y_{\leq t})$ with a tractable fitted model $\hat{f}(X, Y_{\leq t})$, leading to the following approximate future entropy adjustment:

$$\hat{p}^{(\text{ent})}_{\alpha, \hat{f}; X}(Y_t \mid Y_{<t}) \propto \exp \left\{ (1 + \alpha_t) \log \hat{p}_X(Y_t \mid Y_{<t}) - \alpha_t \hat{f}_{t+1}(X, Y_{\leq t}) \right\}. \tag{9}$$

Then, we can initialize $\alpha = (0, ..., 0)$ and first fit $\alpha_T$ for the last generation step. Next, now that the last generation step is calibrated, we can fit future entropy model $\hat{f}_T$, taking in length $T - 1$ prefixes and predicting the entropy at step $T$. Given $\hat{f}_T$, we can then fit $\alpha_{T-1}$, calibrating the second to last step. This procedure proceeds from $t = T, ..., 1$, resulting in a calibrated model.

The future entropy model fitting relies on the following assumption, which states that we can fit regression models that attain good i.i.d. test error:

**Assumption 5.1.** Let $\left( X^{(i)}, Y^{(i)}_{\leq t}, h^{(i)} \right)_{i=1}^n$ be a dataset with inputs $X^{(i)} \overset{\text{i.i.d.}}{\sim} q$ and $Y^{(i)}_{\leq t} \sim p^*_{X^{(i)}}$, along with noisy labels $h^{(i)}$. Furthermore, suppose each noisy label is given by $h^{(i)} = f^*(X^{(i)}, Y^{(i)}_{\leq t}) + \varepsilon_i$ for the true label $f^*(X^{(i)}, Y^{(i)}_{\leq t}) \in \mathbb{R}$ and noise $\varepsilon_i \overset{\text{i.i.d.}}{\sim} \mathcal{E}$, where $\mathcal{E}$ is a mean-zero noise distribution bounded by $\sigma$. Then, there exists a polynomial time algorithm $\mathcal{A}$ which takes in a dataset of size $n \in \text{poly}(\sigma, \delta)$ and outputs a fitted model $\hat{f}$ attaining at most $\delta$ test error:

$$\mathbb{E}_{X \sim q} \mathbb{E}_{Y_{\leq t} \sim p^*_X} |\hat{f}(X, Y_{\leq t}) - f^*(X, Y_{\leq t})| \leq \delta.$$

Empirically, neural networks can fit almost anything while attaining low test error in distribution, and the future entropy prediction problem described here is not particularly complex, involving mapping a set of tokens to a single bounded scalar. However, future empirical work is needed to determine how accurately a large neural model can predict the future entropy.

## 5.3 Main theorem

With this assumption, we now state the main result:

**Theorem 5.2.** *Suppose that Assumption 5.1 holds, where each future entropy predictor attains test error $\delta$. Also, let $(\alpha, \hat{f})$ be the output of Algorithm 1, where each $\alpha_t$ is an $\varepsilon$-stationary point. Then,*

$$\left| EntCE\left( p^* \parallel \hat{p}_{\alpha, \hat{f}}^{(ent)} \right) \right| \leq 2T\delta + \sum_{t=1}^{T}(1 + \alpha_t)\varepsilon,$$

$$\mathcal{L}\left( p^* \parallel \hat{p}_{\alpha, \hat{f}}^{(ent)} \right) \leq \mathcal{L}(p^* \parallel \hat{p}).$$

This theorem tells us that if each future entropy predictor has error $\delta$ and we choose each $\alpha_t$ to be an $\varepsilon$-stationary point with respect to the log loss, the calibrated model will have entropy within $O(\delta + \varepsilon)$ of its log loss at each time step, and its log loss will be better than that of the original model.

*Why does future entropy preserve log loss?* Future entropy adjustment can be derived as a first-order approximation of globally normalized temperature adjustment; we provide this derivation in Appendix B, along with a derivation in the MaxEnt RL framework (Ziebart et al., 2008). Global temperature adjustment attains calibration as long as the gradient of the log loss with respect to temperature is small (Braverman et al., 2020), which is a first-order condition. Then, intuitively, a first-order approximation of global temperature scaling should preserve this property.

The procedure described in Algorithm 1 is not practical to implement, as one would need to a fit a separate future entropy predictor for each generation step and each candidate token, each of which involves a slow data collection process based on a repeated sampling. Nonetheless, the existence of such an algorithm provides evidence that log loss tradeoffs are not inevitable in entropy calibration, despite the output space being exponentially large. One other point to note is that our analysis holds for any approximation of the future entropy that attains error $\delta$, with worse approximations just weakening the calibration error guarantee. For example, one could use the one-step future entropy (Braverman et al., 2020), or truncate to $k$ steps instead. We hope that our theory, which establishes future entropy as the target to approximate, guides future work to achieve better quality-diversity tradeoffs than existing approaches.

## 6 Additional Related Work

Calibration is most commonly studied in binary and multiclass classification, with some classic algorithms including binning, Platt scaling, and isotonic regression (Platt, 1999; Zadrozny & Elkan, 2002; Guo et al., 2017; Kumar et al., 2019). In the language modeling setting, Liang et al. (2023) evaluate the calibration of language models prompted to perform a wide range of classification tasks, finding that models are almost always miscalibrated and overconfident. In such a setting, one can simply apply standard calibration techniques to adjust the model's outputted probabilities. More challenging is linguistic calibration, where models appear overconfident in the language they use to answer a question. To address this problem, past works propose techniques based on controllable generation and reinforcement learning (Mielke et al., 2022; Band et al., 2024). Finally, the term "calibration" is also used to describe the procedure of eliminating the model's innate bias toward certain tokens when doing in-context learning, to improve task performance (Zhao et al., 2021). All of these settings are distinct from our setting, which studies the calibration of a model's entropy over an entire generation, and whose related work we discuss in Section 2.

## 7 Conclusion

We find both theoretically and experimentally that entropy miscalibration improves very slowly with scale. Furthermore, while all current methods calibrate at the cost of diversity, we provide theoretical

evidence that this tradeoff can be avoided. Therefore, given recent community interest in test-time scaling and synthetic data, both for which diversity is centrally important, we are excited about work which seeks to attain both generation stability and diversity simultaneously.

## Acknowledgements

We would like to thank Rishi Bommasani, Sarah Cen, Irena Gao, Konwoo Kim, Suhas Kotha, John Thickstun, and anonymous reviewers for useful conversations about the paper. GV is currently affiliated with OpenAI but did this work while at Stanford. GV and SC were supported by NSF Award AF-2341890 and the Simons Foundation Investigator Award, PL and SC were supported by the Open Philanthropy Project Award, and SC was supported by the NSF Graduate Research Fellowship Program.

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

**Algorithm 3** Future entropy scaling

**Inputs:** model $\hat{p}$, length $T$, vocab $\mathcal{V}$, future entropy fitting algorithm $\mathcal{A}$, future entropy dataset size $n$, sample size $m$, prompt distribution $q$, true conditional distribution $p^*$, optimization tolerate $\varepsilon$

1: Initialize $\alpha_1 = ... = \alpha_T = 0$, $\hat{f}_2 = ... = \hat{f}_{T+1} = 0$.

2: For $t = T, ..., 1$:

3:     Choose $\alpha_t$ to minimize expected log loss at step $t$, until the gradient is at most $\varepsilon$:

$$\alpha_t = \underset{\alpha_t'}{\operatorname{argmin}} \, \mathcal{L}_t \left( p^* \parallel \hat{p}_{\alpha', \hat{f}}^{(\text{ent})} \right)$$

    where $\alpha' = (0, ..., 0, \alpha_t', \alpha_{t+1}, ..., \alpha_T)$.

    ($\mathcal{L}_t$: Equation 3, $\hat{p}_{\alpha', \hat{f}}^{(\text{ent})}$: Equation 9)

4:     Fit the future entropy predictor $\hat{f}_t$ as follows:

5:         Sample prefixes $\left( X^{(i)}, Y_{<t-1}^{(i)} \right)_{i=1}^n$ with $X^{(i)} \overset{\text{i.i.d.}}{\sim} q$, $Y_{<t-1}^{(i)} \sim p_{X^{(i)}}^*$.

6:         For each token $v \in \mathcal{V}$, compute labels $(h^{(i,v)})_{i=1}^n$ by passing each prefix $\left( X^{(i)}, \left[ Y_{<t-1}^{(i)}, v \right] \right)$ into Algorithm 2, along with inputs $\hat{p}_{\alpha, \hat{f}}^{(\text{ent})}$, $T$, $m$.

7:         Fit one future entropy predictor for each token $v$, setting $\hat{f}_t(X, [Y_{<t-1}, v]) = \hat{f}_{t,v}(X, Y_{<t-1})$, where each $\hat{f}_{t,v}$ is the output $\mathcal{A} \left\{ \left( X^{(i)}, Y_{<t-1}^{(i)}, h^{(i,v)} \right)_{i=1}^n \right\}$.

8: Return $(\alpha_1, ..., \alpha_T), (\hat{f}_2, ..., \hat{f}_{T+1})$.

## A   Proofs

Recall notation: we are given prompts $X \in \mathcal{X}$ drawn from some prompt distribution $X \sim q$, and responses $Y \in \mathcal{Y}$ drawn from the true conditional distribution $Y \sim p_X^*$ for $p_X^* \in \Delta^{\mathcal{Y}}$. For simplicity, let $\mathcal{Y}$ be the set $\mathcal{V}^T$ of length $T$ strings over a vocabulary $\mathcal{V}$. We then train a language model $\hat{p} : \mathcal{X} \to \Delta^{\mathcal{Y}}$ to fit the true conditional distribution $p^*$.

We say that $\hat{p}$ is *entropy calibrated* if its entropy over generations is equal to its log loss, in expectation over the prompt:

$$H(\hat{p}) = \mathcal{L}(p^* \parallel \hat{p}), \tag{10}$$

where the total entropy and total log loss are given by

$$H(\hat{p}) = \mathbb{E}_{X \sim q} \mathbb{E}_{\hat{Y} \sim \hat{p}_X}[-\log \hat{p}_X(\hat{Y})], \tag{11}$$

$$\mathcal{L}(p^* \parallel \hat{p}) = \mathbb{E}_{X \sim q} \mathbb{E}_{Y \sim p_X^*}[-\log \hat{p}_X(Y)]. \tag{12}$$

We can also write the per-step entropy and per-step log loss as

$$H_t(\hat{p}) = \mathbb{E}_{X \sim q} \mathbb{E}_{\hat{Y} \sim \hat{p}_X}[-\log \hat{p}_X(\hat{Y}_t \mid \hat{Y}_{<t})], \tag{13}$$

$$\mathcal{L}_t(p^* \parallel \hat{p}) = \mathbb{E}_{X \sim q} \mathbb{E}_{Y \sim p_X^*}[-\log \hat{p}_X(Y_t \mid Y_{<t})]. \tag{14}$$

Let the total entropy calibration error be given by

$$\text{EntCE}(p^* \parallel \hat{p}) = |H(\hat{p}) - \mathcal{L}(\hat{p} \parallel p^*)|$$

$$= \left| \sum_{t=1}^T H_t(\hat{p}) - \mathcal{L}_t(\hat{p} \parallel p^*) \right|. \tag{15}$$

Our goal will be to calibrate the model $\hat{p}$ while preserving its log loss, which we will do by the following adjustment:

$$\hat{p}_{\alpha, \hat{f}; X}^{(\text{ent})}(Y_t \mid Y_{<t}) \propto \exp \left\{ (1 + \alpha_t) \log \hat{p}_X(Y_t \mid Y_{<t}) - \alpha_t \hat{f}_{t+1}(X, Y_{\leq t}) \right\}, \tag{16}$$

where $\alpha_1, ..., \alpha_t$ denote the adjustment parameters, and $\hat{f}_2, ..., \hat{f}_{T+1}$ denote future entropy predictors (with $\hat{f}_{T+1} = 0$), whose goal is to approximate the future entropy. Using Algorithm 3 (copied from Algorithm 1 for convenience) to fit each $\alpha_t, \hat{f}_t$, we show the following result:

**Theorem A.1.** *Suppose that Assumption 5.1 holds, where each future entropy predictor attains test error $\delta$. Also, let $(\alpha, \hat{f})$ be the output of Algorithm 3, where each $\alpha_t$ is an $\varepsilon$-stationary point. Then, we have*

$$\left| EntCE \left( p^* \parallel \hat{p}_{\alpha,\hat{f}}^{(ent)} \right) \right| \leq 2T\delta + \sum_{t=1}^{T} (1 + \alpha_t)\varepsilon,$$

$$\mathcal{L} \left( p^* \parallel \hat{p}_{\alpha,\hat{f}}^{(ent)} \right) \leq \mathcal{L}(p^* \parallel \hat{p}).$$

The proof proceeds as follows: first, recall that for each step $t$, we choose $\alpha_t$ to minimize $\mathcal{L}_t \left( p^* \parallel \hat{p}_{\alpha,\hat{f}}^{(ent)} \right)$. The first lemma will show that if the future entropy predictor $\hat{f}_{t+1}$ fitted in the previous iteration has at most $\delta$ error (in expectation over $Y_{<t}$ and uniformly over $Y_t \in \mathcal{V}$), then this choice of $\alpha_t$ produces a calibration-like guarantee.

**Lemma A.2.** *Suppose that $\alpha_t$ is an $\varepsilon$-stationary point with respect to $\mathcal{L}_t$:*

$$\left| \frac{d}{d\alpha_t} \mathcal{L}_t \left( p^* \parallel \hat{p}_{\alpha,\hat{f}}^{(ent)} \right) \right| \leq \varepsilon,$$

*and that the future entropy predictor $\hat{f}_{t+1}$ attains at most $\delta$ error, in expectation over $Y_{<t}$ and uniformly over $Y_t \in \mathcal{V}$:*

$$\max_{Y_t \in V} \mathbb{E}_{X \sim q} \mathbb{E}_{Y_{<t} \sim p_X^*} \left| \hat{f}_{t+1}(X, Y_{\leq t}) - H_{\hat{p}_{\alpha,\hat{f};X}^{(ent)}} (Y_{>t} \mid Y_{\leq t}) \right| \leq \delta.$$

*Then, we have the following calibration guarantee:*

$$\left| \mathbb{E}_{X \sim q} \mathbb{E}_{Y_{\leq t} \sim p_X^*} \mathbb{E}_{\hat{Y}_{>t} \sim \hat{p}_{\alpha,\hat{f};X}^{(ent)}(\cdot|Y_{\leq t})} \left[ -\log \hat{p}_{\alpha,\hat{f};X}^{(ent)} (Y_{\leq t}, \hat{Y}_{>t}) \right] \right.$$

$$\left. - \mathbb{E}_{X \sim q} \mathbb{E}_{Y_{<t} \sim p_X^*} \mathbb{E}_{\hat{Y}_{\geq t} \sim \hat{p}_{\alpha,\hat{f};X}^{(ent)}(\cdot|Y_{<t})} \left[ -\log \hat{p}_{\alpha,\hat{f};X}^{(ent)} (Y_{<t}, \hat{Y}_{\geq t}) \right] \right| \leq (1 + \alpha_t)\varepsilon + 2\delta.$$

This bound can be thought of as a partial calibration guarantee in the sense that it allows us to swap $Y_t \sim p^*$ and $\hat{Y}_t \sim \hat{p}_{\alpha,\hat{f}}^{(ent)}$ in the expectation.

To show that Algorithm 3 improves log loss, note that each $\alpha_t$ is initialized to 0, so the initial model $\hat{p}_{\alpha,\hat{f}}^{(ent)}$ is equal to $\hat{p}$. Then, it suffices to show that each iteration of the algorithm improves the log loss, relative to the previous iteration. This statement is true by the following lemma, which states that at each step $t$ in the algorithm, optimizing $\mathcal{L}_t$ is equivalent to optimizing the overall log loss $\mathcal{L}$:

**Lemma A.3.** *Let $\alpha_{t+1}, ..., \alpha_T$ be set arbitrarily, and let $\alpha_1 = ... = \alpha_{t-1} = 0$. Also, let $\hat{f}$ be set arbitrarily. Then,*

$$\underset{\alpha_t'}{\operatorname{argmin}} \, \mathcal{L}_t \left( p^* \parallel \hat{p}_{\alpha',\hat{f}}^{(ent)} \right) = \underset{\alpha_t'}{\operatorname{argmin}} \, \mathcal{L} \left( p^* \parallel \hat{p}_{\alpha',\hat{f}}^{(ent)} \right),$$

*where $\alpha' = (0, ..., 0, \alpha_t', \alpha_{t+1}, ..., \alpha_T)$.*

The final lemma involves showing that each future entropy predictor outputted by the algorithm attains low error and satisfies the condition in Lemma A.2. This lemma relies on the fact that the future entropy $H_{\hat{p}_{\alpha,\hat{f};X}^{(ent)}} (Y_{>t} \mid Y_{\leq t})$ only depends on $\alpha_{t+1}, ..., \alpha_T$ and $\hat{f}_{t+2}, ..., \hat{f}_{T+1}$, because it only involves generation steps $t + 1$ and onward. Therefore, after $\alpha_{t+1}$ is chosen, the generation process is fixed for steps $t + 1$ and onward, so we can fit a future entropy predictor over those steps despite not having yet chosen $\alpha_1, ..., \alpha_t$. These facts, along with the black box fitting procedure provided in Assumption 5.1, lead to the following lemma:

**Lemma A.4.** *For any $\alpha = (\alpha_1, ..., \alpha_T)$ and $\hat{f} = (\hat{f}_2, ..., \hat{f}_{T+1})$, and for some fixed $t$, let $\alpha' = (0, ..., 0, \alpha_t, ..., \alpha_T)$ and $\hat{f}' = (0, ..., 0, \hat{f}_{t+1}, ..., \hat{f}_{T+1})$ be the results of zeroing out the first $t-1$ entries of $\alpha$ and $\hat{f}$. Then, we have that*

$$H_{\hat{p}^{(ent)}_{\alpha, \hat{f}; X}}(Y_{>t-1} \mid Y_{\leq t-1}) = H_{\hat{p}^{(ent)}_{\alpha', \hat{f}'; X}}(Y_{>t-1} \mid Y_{\leq t-1})$$

*for all $Y_{\leq t-1}$. Furthermore, suppose that Assumption 5.1 holds, and let $\mathcal{D} = \left(X^{(i)}, Y^{(i)}_{<t-1}, h^{(i,v)}\right)^n_{i=1}$ be a dataset with*

$$h^{(i,v)} = H_{\hat{p}^{(ent)}_{\alpha', \hat{f}'; X}}\left(Y_{>t-1} \mid \left[Y^{(i)}_{<t-1}, v\right]\right) + \varepsilon_{i,v}$$

$$X^{(i)} \overset{i.i.d.}{\sim} q, \ Y^{(i)}_{<t-1} \sim p^*_{X^{(i)}}, \varepsilon_{i,v} \sim \mathcal{E}$$

*for some token $v \in \mathcal{V}$, dataset size $n = \mathrm{poly}(T \log \mathcal{V}, \delta)$, and some mean-zero noise distribution $\mathcal{E}$ bounded by $T \log \mathcal{V}$. Then, letting $\mathcal{A}$ denote the black box fitting procedure in Assumption 5.1, we have that $\hat{f}_{t,v} = \mathcal{A}(\mathcal{D})$ satisfies*

$$\mathbb{E}_{X \sim q} \mathbb{E}_{Y_{<t-1} \sim p^*_X} \left| \hat{f}_{t,v}(X, Y_{<t-1}) - H_{\hat{p}^{(ent)}_{\alpha, \hat{f}; X}}(Y_{>t-1} \mid [Y_{<t-1}, v]) \right| \leq \delta.$$

We use these lemmas to prove Theorem A.1 as follows:

*Proof of Theorem A.1.* Let $\alpha = (\alpha_1, ..., \alpha_T)$ and $\hat{f} = (\hat{f}_2, ..., \hat{f}_{T+1})$ denote the outputs of the algorithm. It suffices to show the following three inequalities for all $t$:

(a) Prediction error bound: the predictor $\hat{f}_{t+1}$ satisfies

$$\max_{Y_t \in V} \mathbb{E}_{X \sim q} \mathbb{E}_{Y_{<t} \sim p^*_X} \left| \hat{f}_{t+1}(X, Y_{\leq t}) - H_{\hat{p}^{(ent)}_{\alpha, \hat{f}; X}}(Y_{>t} \mid Y_{\leq t}) \right| \leq \delta.$$

(b) Calibration bound: after iteration $t$ of the algorithm, we have

$$\left| \mathbb{E}_{X \sim q} \mathbb{E}_{Y_{\leq t} \sim p^*_X} \mathbb{E}_{\hat{Y}_{>t} \sim \hat{p}^{(ent)}_{\alpha, \hat{f}; X}(\cdot | Y_{\leq t})} \left[ -\log \hat{p}^{(ent)}_{\alpha, \hat{f}; X}(Y_{\leq t}, \hat{Y}_{>t}) \right] \right.$$

$$\left. - \mathbb{E}_{X \sim q} \mathbb{E}_{Y_{<t} \sim p^*_X} \mathbb{E}_{\hat{Y}_{\geq t} \sim \hat{p}^{(ent)}_{\alpha, \hat{f}; X}(\cdot | Y_{<t})} \left[ -\log \hat{p}^{(ent)}_{\alpha, \hat{f}; X}(Y_{<t}, \hat{Y}_{\geq t}) \right] \right| \leq (1 + \alpha_t)\varepsilon + 2\delta.$$

(c) Log loss improvement: letting $\alpha^{(t)} = (0, ..., 0, \alpha_t, ..., \alpha_T)$ and $\hat{f}^{(t)} = (0, ..., 0, \hat{f}_{t+1}, ..., \hat{f}_{T+1})$, we have

$$\mathcal{L}\left(p^* \parallel \hat{p}^{(ent)}_{\alpha^{(t)}, \hat{f}^{(t)}}\right) \leq \mathcal{L}\left(p^* \parallel \hat{p}^{(ent)}_{\alpha^{(t+1)}, \hat{f}^{(t+1)}}\right).$$

The theorem follows from combining these inequalities for all $t$: first, to show that log loss improves, it suffices to apply inequality (c) (log loss improvement) for all $t$, where $\hat{p}^{(ent)}_{\alpha^{(1)}, \hat{f}^{(1)}} = \hat{p}^{(ent)}_{\alpha, \hat{f}}$ and $\hat{p}^{(ent)}_{\alpha^{(T+1)}, \hat{f}^{(T+1)}} = \hat{p}$. Similarly, the calibration result follows from applying inequality (b) (calibration bound) for all $t$ with triangle inequality:

$$\left| \mathrm{EntCE}\left(p^* \parallel \hat{p}^{(ent)}_{\alpha, \hat{f}}\right) \right| = \left| \mathbb{E}_{X \sim q} \mathbb{E}_{Y \sim p^*_X} \left[ -\log \hat{p}^{(ent)}_{\alpha, \hat{f}; X}(Y) \right] - \mathbb{E}_{X \sim q} \mathbb{E}_{Y \sim \hat{p}^{(ent)}_{\alpha, \hat{f}; X}(Y)} \left[ -\log \hat{p}^{(ent)}_{\alpha, \hat{f}; X}(Y) \right] \right|$$

$$= \left| \sum_{t=1}^T \mathbb{E}_{X \sim q} \mathbb{E}_{Y_{\leq t} \sim p^*_X} \mathbb{E}_{\hat{Y}_{>t} \sim \hat{p}^{(ent)}_{\alpha, \hat{f}; X}(\cdot | Y_{\leq t})} \left[ -\log \hat{p}^{(ent)}_{\alpha, \hat{f}; X}(Y_{\leq t}, \hat{Y}_{>t}) \right] \right.$$

$$\left. - \mathbb{E}_{X \sim q} \mathbb{E}_{Y_{<t} \sim p^*_X} \mathbb{E}_{\hat{Y}_{\geq t} \sim \hat{p}^{(ent)}_{\alpha, \hat{f}; X}(\cdot | Y_{<t})} \left[ -\log \hat{p}^{(ent)}_{\alpha, \hat{f}; X}(Y_{<t}, \hat{Y}_{\geq t}) \right] \right|$$

$$\leq \sum_{t=1}^T [(1 + \alpha_t)\varepsilon + 2\delta].$$

Then, showing inequalities (a), (b), and (c) for all $t$ completes the proof. First, note that if inequality (a) (prediction error bound) holds for all $t$, then the other two inequalities follow directly from the lemmas: inequality (a) ensures the condition in Lemma A.2 is satisfied, directly proving inequality (b) (calibration bound). Inequality (c) (log loss improvement) follows from the fact that $\alpha_T$ is chosen via $\operatorname{argmin}_{\alpha'_T} \mathcal{L}_T \left( p^* \parallel \hat{p}_{\alpha',\hat{f}}^{(\text{ent})} \right)$, which by Lemma A.3 is equivalent to minimizing the overall log loss $\mathcal{L} \left( p^* \parallel \hat{p}_{\alpha',\hat{f}}^{(\text{ent})} \right)$.

To show inequality (a) (prediction error bound), first note that for $t = T$, it holds trivially because the future entropy is 0. For $t = 1, ..., T - 1$, the prediction error bound follows directly from applying Lemma A.4 for each $\hat{f}_{t+1,v}$ for $v \in \mathcal{V}$, where each noisy future entropy label computed via parallel sampling (Algorithm 2) has mean equal to the future entropy and is bounded by $(T - t) \log \mathcal{V}$. $\square$

The proofs of the three lemmas proceed as follows:

*Proof of Lemma A.2.* Taking the derivative of the log loss $\mathcal{L}_t$ with respect to $\alpha_t$, we have

$$
\begin{aligned}
\varepsilon &\geq \left| \frac{d}{d\alpha_t} \mathcal{L}_t \left( p^* \parallel \hat{p}_{\alpha,\hat{f}}^{(\text{ent})} \right) \right| \\
&= \left| \frac{d}{d\alpha_t} \mathbb{E}_{X \sim q} \mathbb{E}_{Y \sim p_X^*} [-\mathbb{1}_{Y_t}(\cdot)^T \log \operatorname{softmax}((1 + \alpha_t) \log \hat{p}_X(\cdot \mid Y_{<t}) - \alpha_t \hat{f}_{t+1}(X, [Y_{<t}, \cdot]))] \right| \\
&= \left| \mathbb{E}_{X \sim q} \mathbb{E}_{Y \sim p_X^*} \left[ - \left( \mathbb{1}_{Y_t}(\cdot) - \hat{p}_{\alpha,\hat{f};X}^{(\text{ent})}(\cdot \mid Y_{<t}) \right)^T (\log \hat{p}_X(\cdot \mid Y_{<t}) - \hat{f}_{t+1}(X, [Y_{<t}, \cdot])) \right] \right|,
\end{aligned}
$$

where we use $f(\cdot) \in \mathbb{R}^{|\mathcal{V}|}$ to denote the vector $[f(v)]_{v \in \mathcal{V}}$, the indicator function is given by $\mathbb{1}_{Y_t}(v) = 1$ iff $Y_t = v$, and $\operatorname{softmax} : \mathbb{R}^{|\mathcal{V}|} \to \mathbb{R}^{|\mathcal{V}|}$ applies the softmax operation, which exponentiates each entry and then normalizes the vector by its sum. Splitting this term into two expectations results in the expression

$$
= \left| \mathbb{E}_{X \sim q} \mathbb{E}_{Y_{\leq t} \sim p_X^*} \left[ -(\log \hat{p}_X(Y_t \mid Y_{<t}) - \hat{f}_{t+1}(X, Y_{\leq t})) \right] \right.
$$

$$
\left. - \mathbb{E}_{X \sim q} \mathbb{E}_{Y_{<t} \sim p_X^*} \mathbb{E}_{\hat{Y}_t \sim \hat{p}_{\alpha,\hat{f};X}^{(\text{ent})}(\cdot \mid Y_{<t})} \left[ -(\log \hat{p}_X(\hat{Y}_t \mid Y_{<t}) - \hat{f}_{t+1}(X, [Y_{<t}, \hat{Y}_t])) \right] \right|,
$$

where the two terms differ in whether $Y_t \sim p^*$ or $\hat{Y}_t \sim \hat{p}_{\alpha,\hat{f};X}^{(\text{ent})}$. Multiplying both sides by $(1 + \alpha_t)$, we have

$$
(1 + \alpha_t)\varepsilon \geq \left| \mathbb{E}_{X \sim q} \mathbb{E}_{Y_{\leq t} \sim p_X^*} \left[ -(1 + \alpha_t)(\log \hat{p}_X(Y_t \mid Y_{<t}) - \hat{f}_{t+1}(X, Y_{\leq t})) \right] \right.
$$

$$
\left. - \mathbb{E}_{X \sim q} \mathbb{E}_{Y_{<t} \sim p_X^*} \mathbb{E}_{\hat{Y}_t \sim \hat{p}_{\alpha,\hat{f};X}^{(\text{ent})}(\cdot \mid Y_{<t})} \left[ -(1 + \alpha_t)(\log \hat{p}_X(\hat{Y}_t \mid Y_{<t}) - \hat{f}_{t+1}(X, [Y_{<t}, \hat{Y}_t])) \right] \right|.
$$

Next, noticing that both expressions include unnormalized logits for the distribution $p_{\alpha,\hat{f};X}^{(\text{ent})}$ applied to either $Y_t$ or $\hat{Y}_t$, we can subtract the same normalizing constant from both expressions, resulting in

$$
= \left| \mathbb{E}_{X \sim q} \mathbb{E}_{Y_{\leq t} \sim p_X^*} \left[ - \left( \log \hat{p}_{\alpha,\hat{f};X}^{(\text{ent})}(Y_t \mid Y_{<t}) - \hat{f}_{t+1}(X, Y_{\leq t}) \right) \right] \right.
$$

$$
\left. - \mathbb{E}_{X \sim q} \mathbb{E}_{Y_{<t} \sim p_X^*} \mathbb{E}_{\hat{Y}_t \sim \hat{p}_{\alpha,\hat{f};X}^{(\text{ent})}(\cdot \mid Y_{<t})} \left[ - \left( \log \hat{p}_{\alpha,\hat{f};X}^{(\text{ent})}(\hat{Y}_t \mid Y_{<t}) - \hat{f}_{t+1}(X, [Y_{<t}, \hat{Y}_t]) \right) \right] \right|.
$$

Next, to turn each conditional probability into a joint probability, we can add $\mathbb{E}_{X \sim q} \mathbb{E}_{Y_{\leq t} \sim p_X^*} \left[ -\log \hat{p}_{\alpha, \hat{f}; X}^{(\text{ent})}(Y_{<t}) \right]$ to both expressions:

$$= \left| \mathbb{E}_{X \sim q} \mathbb{E}_{Y_{\leq t} \sim p_X^*} \left[ -\left( \log \hat{p}_{\alpha, \hat{f}; X}^{(\text{ent})}(Y_t, Y_{<t}) - \hat{f}_{t+1}(X, Y_{\leq t}) \right) \right] \right.$$

$$\left. - \mathbb{E}_{X \sim q} \mathbb{E}_{Y_{<t} \sim p_X^*} \mathbb{E}_{\hat{Y}_t \sim \hat{p}_{\alpha, \hat{f}; X}^{(\text{ent})}(\cdot | Y_{<t})} \left[ -\left( \log \hat{p}_{\alpha, \hat{f}; X}^{(\text{ent})}(\hat{Y}_t, Y_{<t}) - \hat{f}_{t+1}(X, [Y_{<t}, \hat{Y}_t]) \right) \right] \right|.$$

At this point, we can use the fact that $\hat{f}_{t+1}$ is within $\delta$ of the future entropy (in expectation over $X \sim q$, $Y_{<t} \sim p_X^*$ and uniformly over $Y_t$) to produce the bound

$$(1 + \alpha_t)\varepsilon + 2\delta \geq \left| \mathbb{E}_{X \sim q} \mathbb{E}_{Y_{\leq t} \sim p_X^*} \left[ -\left( \log \hat{p}_{\alpha, \hat{f}; X}^{(\text{ent})}(Y_t, Y_{<t}) - H_{\hat{p}_{\alpha, \hat{f}; X}^{(\text{ent})}}(Y_{>t} \mid Y_{\leq t}) \right) \right] \right.$$

$$\left. - \mathbb{E}_{X \sim q} \mathbb{E}_{Y_{<t} \sim p_X^*} \mathbb{E}_{\hat{Y}_t \sim \hat{p}_{\alpha, \hat{f}; X}^{(\text{ent})}(\cdot | Y_{<t})} \left[ -\left( \log \hat{p}_{\alpha, \hat{f}; X}^{(\text{ent})}(\hat{Y}_t, Y_{<t}) - H_{\hat{p}_{\alpha, \hat{f}; X}^{(\text{ent})}}(Y_{>t} \mid [Y_{<t}, \hat{Y}_t]) \right) \right] \right|.$$

Finally, note that the future entropy is defined as

$$H_{\hat{p}_{\alpha, \hat{f}; X}^{(\text{ent})}}(Y_{>t} \mid Y_{\leq t}) = \mathbb{E}_{\hat{Y}_{>t} \sim \hat{p}_{\alpha, \hat{f}; X}^{(\text{ent})}(\cdot | Y_{\leq t})} \left[ -\log \hat{p}_{\alpha, \hat{f}; X}^{(\text{ent})}(\hat{Y}_{>t} \mid Y_{\leq t}) \right],$$

which we can substitute into the previous equation to produce the desired result:

$$(1 + \alpha_t)\varepsilon + 2\delta \geq \left| \mathbb{E}_{X \sim q} \mathbb{E}_{Y_{\leq t} \sim p_X^*} \mathbb{E}_{\hat{Y}_{>t} \sim \hat{p}_{\alpha, \hat{f}; X}^{(\text{ent})}(\cdot | Y_{\leq t})} \left[ -\log \hat{p}_{\alpha, \hat{f}; X}^{(\text{ent})}(\hat{Y}_{>t}, Y_t, Y_{<t}) \right] \right.$$

$$\left. - \mathbb{E}_{X \sim q} \mathbb{E}_{Y_{<t} \sim p_X^*} \mathbb{E}_{\hat{Y}_{\geq t} \sim \hat{p}_{\alpha, \hat{f}; X}^{(\text{ent})}(\cdot | Y_{<t})} \left[ -\log \hat{p}_{\alpha, \hat{f}; X}^{(\text{ent})}(\hat{Y}_{>t}, \hat{Y}_t, Y_{<t}) \right] \right|.$$

$\square$

*Proof of Lemma A.3.* Let $t$ denote the time step of interest. Writing the full log loss as a sum over $s$, we have

$$\mathcal{L}\left( p^* \parallel \hat{p}_{\alpha, \hat{f}}^{(\text{ent})} \right) = \sum_{s=1}^{T} \mathcal{L}_s \left( p^* \parallel \hat{p}_{\alpha, \hat{f}}^{(\text{ent})} \right).$$

By the definition of $\hat{p}_{\alpha, \hat{f}}^{(\text{ent})}$, the $t$-th parameter $\alpha_t$ has no effect on summands $s \neq t$. Therefore, optimizing the entire sum is equivalent to optimizing only the summand corresponding to $s = t$, proving the desired result. $\square$

*Proof of Lemma A.4.* First, to show that

$$H_{\hat{p}_{\alpha, \hat{f}; X}^{(\text{ent})}}(Y_{>t-1} \mid Y_{\leq t-1}) = H_{\hat{p}_{\alpha', \hat{f}'; X}^{(\text{ent})}}(Y_{>t-1} \mid Y_{\leq t-1})$$

where $\alpha'$, $\hat{f}'$ are the results of zeroing out the first $t-1$ entries of $\alpha$, $\hat{f}$, we can simply write out the definition of the future entropy:

$$H_{\hat{p}_{\alpha, \hat{f}; X}^{(\text{ent})}}(Y_{>t-1} \mid Y_{\leq t-1}) = \mathbb{E}_{\hat{Y}_{>t-1} \sim \hat{p}_{\alpha, \hat{f}; X}^{(\text{ent})}(\cdot | Y_{\leq t-1})} \left[ -\log \hat{p}_{\alpha, \hat{f}; X}^{(\text{ent})}(\hat{Y}_{>t-1} \mid Y_{\leq t-1}) \right],$$

where we can write out the probability as

$$\hat{p}_{\alpha, \hat{f}; X}^{(\text{ent})}(\hat{Y}_{>t-1} \mid Y_{\leq t-1}) = \prod_{s=t}^{T} \hat{p}_{\alpha, \hat{f}; X}^{(\text{ent})}(\hat{Y}_s \mid Y_{\leq t-1}, \hat{Y}_{t,\ldots,s-1})$$

$$= \prod_{s=t}^{T} \mathbb{1}_{\hat{Y}_s}^{T} \operatorname{softmax}\left( (1 + \alpha_s) \log \hat{p}_X(\cdot \mid Y_{\leq t-1}, \hat{Y}_{t,\ldots,s-1}) \right.$$

$$\left. - \alpha_s \hat{f}_{s+1}(X, [Y_{\leq t-1}, \hat{Y}_{t,\ldots,s-1}, \cdot]) \right).$$

This expression has no dependence on the first $t-1$ entries $\alpha_1, ..., \alpha_{t-1}$ of $\alpha$, and no dependence on the first $t-1$ entries $\hat{f}_2, ..., \hat{f}_t$ of $\hat{f}$, proving the first half of the lemma.

The second half of the lemma follows directly from applying Assumption 5.1, where $\alpha'$, $\hat{f}'$ and $\alpha$, $\hat{f}$ can be interchanged by the fact that their future entropies over steps $t, ..., T$ are the same. $\qquad\square$

## B  Derivations

### B.1  Entropy calibration from binary calibration

Recall that for a binary classifier $\hat{f} : \mathcal{X} \to [0, 1]$, where $f^* : \mathcal{X} \to [0, 1]$ denotes the true conditional distribution, binary calibration asks whether the model's probability corresponds to the actual fraction of ones in reality:

$$\mathbb{E}_{X \sim q} \mathbb{E}_{Y \sim f_X^*} [Y \mid \hat{f}_X = p] = p.$$

First, note that the right hand side can be replaced by

$$\mathbb{E}_{X \sim q} \mathbb{E}_{Y \sim f_X^*} [Y \mid \hat{f}_X = p] = \mathbb{E}_{X \sim q} \mathbb{E}_{\hat{Y} \sim \hat{f}_X} [\hat{Y} \mid \hat{f}_X = p].$$

Next, we can weaken this requirement by making the expectation non-conditional, or

$$\mathbb{E}_{X \sim q} \mathbb{E}_{Y \sim f_X^*} Y = \mathbb{E}_{X \sim q} \mathbb{E}_{\hat{Y} \sim \hat{f}_X} \hat{Y},$$

which simply asks whether the overall rate of ones under $\hat{f}$ is the same as the overall rate of ones in reality. The most natural extension of this definition to multiclass calibration is top-class calibration,

$$\mathbb{E}_{X \sim q} \mathbb{E}_{Y \sim f_X^*} \left[ \mathbb{1} \left\{ Y = \operatorname*{argmax}_{y'} \hat{f}_X(y') \right\} \mid \max_{y'} \hat{f}_X(y') = p \right] = p,$$

which states that across all instances where the model assigns $p$ probability to the top class, the actual label should be equal to the top class $p$ fraction of the time on average. Like before, we can replace the right hand side by

$$\mathbb{E}_{X \sim q} \mathbb{E}_{Y \sim f_X^*} \left[ \mathbb{1} \left\{ Y = \operatorname*{argmax}_{y'} \hat{f}_X(y') \right\} \mid \max_{y'} \hat{f}_X(y') = p \right]$$
$$= \mathbb{E}_{X \sim q} \mathbb{E}_{\hat{Y} \sim \hat{f}_X} \left[ \mathbb{1} \left\{ \hat{Y} = \operatorname*{argmax}_{y'} \hat{f}_X(y') \right\} \mid \max_{y'} \hat{f}_X(y') = p \right],$$

where $Y \sim f_X^*$ and $\hat{Y} \sim \hat{f}_X$ are interchanged. In this expression, the top class probability $\max_{y'} \hat{f}_X(y')$ can be thought of as the confidence of $\hat{f}_X$, while the zero-one loss function $\mathbb{1} \left\{ \hat{Y} = \operatorname*{argmax}_{y'} \hat{f}_X(y') \right\}$ defines the metric the confidence should be calibrated to — the model's confidence should correspond to the loss it incurs in reality. For language models, it is natural to replace the zero-one loss with the log loss, which produces the definition

$$\mathbb{E}_{X \sim q} \mathbb{E}_{Y \sim f_X^*} \left[ -\log \hat{f}_X(Y) \mid H(f_X) = h \right] = h$$
$$= \mathbb{E}_{X \sim q} \mathbb{E}_{\hat{Y} \sim \hat{f}_X} \left[ -\log \hat{f}_X(\hat{Y}) \mid H(f_X) = h \right],$$

which asks whether the model's entropy corresponds to the log loss it incurs in reality. We study the unconditional version of this definition

$$\mathbb{E}_{X \sim q} \mathbb{E}_{Y \sim f_X^*} \left[ -\log \hat{f}_X(Y) \right] = \mathbb{E}_{X \sim q} \mathbb{E}_{\hat{Y} \sim \hat{f}_X} \left[ -\log \hat{f}_X(\hat{Y}) \right],$$

which simply asks whether the model's entropy matches its log loss on average. We study unconditional calibration for simplicity, but the same techniques to calibrate unconditionally would likely work for conditional calibration as well if one buckets the inputs $X$ by their entropy $H(f_X)$.

## B.2 Future entropy adjustment from global temperature adjustment

To derive future entropy adjustment from global temperature adjustment, recall that the global temperature adjustment with respect to inverse temperature $\alpha$ (where $\tau = 1/(1+\alpha)$) is given by

$$p_\alpha^{(\text{global})}(Y_1, ..., Y_T) = \frac{p(Y_1, ..., Y_T)^{1+\alpha}}{\sum_{Y' \in \mathcal{V}^T} p(Y_1', ..., Y_T')^{1+\alpha}}.$$

Factoring this joint distribution into a conditional distribution for each $t$, we have

$$\log p_\alpha^{(\text{global})}(Y_t \mid Y_{<t}) = \log \frac{\sum_{Y_{>t}} p(Y_{<t}, Y_t, Y_{>t})^{1+\alpha}}{\sum_{Y_t', Y_{>t}} p(Y_{<t}, Y_t', Y_{>t})^{1+\alpha}}.$$

Taking the gradient of the log probability with respect to $\alpha$, we have

$$\frac{d}{d\alpha} \log p_\alpha^{(\text{global})}(Y_t \mid Y_{<t}) = \text{softmax} \left\{ (1+\alpha) \log p(Y_{<t}, Y_t, Y_{>t} = \cdot) \right\}^T \log p(Y_{<t}, Y_t, Y_{>t} = \cdot)$$
$$- \text{softmax} \left\{ (1+\alpha) \log p(Y_{<t}, [Y_t, Y_{>t}] = \cdot) \right\}^T \log p(Y_{<t}, [Y_t, Y_{>t}] = \cdot),$$

where the first softmax is over $Y_{>t}$ and the second softmax is over both $Y_t$ and $Y_{>t}$. Simplifying this expression results in

$$= \log p(Y_t \mid Y_{<t}) + \mathbb{E}_{Y_{>t} \sim p_\alpha^{(\text{global})}(\cdot | Y_{\le t})} \log p(Y_{>t} \mid Y_{\le t}) - \mathbb{E}_{Y_{\ge t} \sim p_\alpha^{(\text{global})}(\cdot | Y_{<t})} \log p(Y_{\ge t} \mid Y_{<t}).$$

Then, the first-order approximation of $\log p_\alpha^{(\text{global})}(Y_t \mid Y_{<t})$ centered around $\alpha = 0$ is given by

$$\log p_\alpha^{(\text{global})}(Y_t \mid Y_{<t}) \approx \log p_{\alpha=0}^{(\text{global})}(Y_t \mid Y_{<t}) + \alpha \frac{d}{d\alpha} \log p_\alpha^{(\text{global})}(Y_t \mid Y_{<t}) \Bigg|_{\alpha=0}$$

$$= \log p(Y_t \mid Y_{<t})$$
$$+ \alpha \Bigg[ \log p(Y_t \mid Y_{<t}) + \mathbb{E}_{Y_{>t} \sim p_{\alpha=0}^{(\text{global})}(\cdot | Y_{\le t})} \log p(Y_{>t} \mid Y_{\le t})$$
$$- \mathbb{E}_{Y_{\ge t} \sim p_{\alpha=0}^{(\text{global})}(\cdot | Y_{<t})} \log p(Y_{\ge t} \mid Y_{<t}) \Bigg]$$
$$= (1+\alpha) \log p(Y_t \mid Y_{<t}) - \alpha \mathbb{E}_{Y_{>t} \sim p(\cdot | Y_{\le t})}[-\log p(Y_{>t} \mid Y_{\le t})] + C_{Y_{\le t}},$$

where the final term is constant with respect to $Y_t$.

## B.3 Future entropy adjustment from MaxEnt RL

The future entropy adjustment can also be derived in the MaxEnt RL framework (Ziebart et al., 2008), where the reward function is given by $r(x, y) = \log \hat{p}_x(y)$ with $\hat{p}$ denoting the base model. Specifically, we can write the MaxEnt RL objective as

$$\max_{\tilde{p}} \mathbb{E}_{X \sim q} \mathbb{E}_{Y \sim \tilde{p}_X} r_X(Y) - \alpha \text{KL}(\tilde{p} \parallel \hat{p}).$$

Then, the value function for this objective is given by

$$V_X(Y_{\le t}) = \mathbb{E}_{Y_{>t} \sim \tilde{p}_X(Y_{>t} | Y_{\le t})} \log \hat{p}_X(Y_{>t} \mid Y_{\le t}),$$

and the Q function is given by

$$Q_X(Y_{<t}, Y_t) = r_X(Y_t \mid Y_{<t}) + V_X(Y_{\le t})$$
$$= \log \hat{p}_X(Y_t \mid Y_{<t}) + \mathbb{E}_{Y_{>t} \sim \tilde{p}_X(Y_{>t} | Y_{\le t})} \log \hat{p}_X(Y_{>t} \mid Y_{\le t}).$$

Using this Q function to define the KL-regularized policy then results in

$$\tilde{p}_{\alpha;X}(Y_t \mid Y_{<t}) \propto \exp \left\{ \log \hat{p}_X(Y_t \mid Y_{<t}) + \alpha Q_X(Y_{<t}, Y_t) \right\}$$
$$= \exp \left\{ (1+\alpha) \log \hat{p}_X(Y_t \mid Y_{<t}) - \alpha \mathbb{E}_{Y_{>t} \sim \tilde{p}_{\alpha;X}(Y_{>t} | Y_{\le t})}[-\log \hat{p}_X(Y_{>t} \mid Y_{\le t})] \right\},$$

which is the future entropy adjustment.

### B.4 Scaling in the simplified setting

Recall our simplified setup: the model sees $m$ tokens drawn i.i.d. from an $\alpha$ power law distribution over a vocabulary of size $v$, and it stores the count of each token it sees. At generation time, the model generates a sequence of length $L$ as follows: if the context contains only tokens the model has seen more than once, it behaves normally and produces the next token according to its fitted unigram distribution. But if the context contains at least one token that the model saw only once, then it instead produces the next tokens according to some derailed distribution with entropy larger by some constant $C_H$.

First, if the per-step derailing probability $q$ is small, we can compute expected entropy at time $t$ as follows using the binomial approximation:

$$\begin{aligned}
H_t(\hat{p}) &= (1-q)^t H_0 + (1 - (1-q)^t)(H_0 + C_H) \\
&\approx (1 - qt)H_0 + (1 - (1 - qt))(H_0 + C_H) \\
&= H_0 + qt C_H,
\end{aligned}$$

so the overall miscalibration is given by

$$\begin{aligned}
\sum_{t=1}^{L} H_t(\hat{p}) - H_0 &= \sum_{t=1}^{L} qt C_H \\
&= q C_H \frac{L(L-1)}{2}.
\end{aligned}$$

Next, to characterize the scaling of the expected per-step derailing probability $q$ with respect to dataset size $m$, we first note that

$$q = \frac{K_{m,1}}{m},$$

where $K_{m,1}$ is a random variable denoting the number of items seen exactly once in the training set of size $m$. Taking the expectation with respect to random draws of the training set, we have

$$\begin{aligned}
\mathbb{E} K_{m,1} &= \mathbb{E} \sum_{i=1}^{v} \mathbb{1}\{\text{count}_m(i) = 1\} \\
&= \sum_{i=1}^{v} \mathbb{E} \mathbb{1}\{\text{count}_m(i) = 1\} \\
&= \sum_{i=1}^{v} m p_i (1 - p_i)^{m-1},
\end{aligned}$$

where $p_i = Z/i^\alpha$ is the power law probability of the $i$th item, with $Z = \sum_{i=1}^{v} 1/i^\alpha$ denoting the normalizing constant. Next, taking $v \to \infty$ following the infinite urn setup in Good (1953); Karlin (1967), we compute

$$\begin{aligned}
\int_{i=1}^{\infty} m p_i (1 - p_i)^{m-1} di &= \int_{i=1}^{\infty} m Z i^{-\alpha} (1 - Z i^{-\alpha})^{m-1} di \\
&= \frac{1}{\alpha} Z^{\frac{1}{\alpha}} (m-1)^{\frac{1}{\alpha}} \gamma(1 - 1/\alpha, (m-1)Z),
\end{aligned}$$

where

$$\gamma(a, x) = \int_0^x t^{a-1} e^{-t} dt$$

is the lower incomplete gamma function. Taking $m \to \infty$ and using the fact that $\gamma(a, x) \to \Gamma(a)$ for $x \to \infty$, we have that

$$\mathbb{E} \frac{K_{m,1}}{m} \sim \frac{1}{\alpha} Z^{\frac{1}{\alpha}} m^{\frac{1}{\alpha} - 1} \Gamma(1 - 1/\alpha),$$

as desired. This expression can also be found in Equation 23 of Karlin (1967).

## C  Experimental details

We study four model families (**Qwen2.5** (0.5B, 1.5B, 3B, 7B, 14B, 32B, 72B) (Qwen et al., 2025), **Llama 3** (1B, 3B, 8B, 70B) (Grattafiori et al., 2024), **Llama 2** (7B, 13B, 70B) (Touvron et al., 2023), and **Pythia** (410M, 1.4B, 2.8B, 6.9B, 12B) (Biderman et al., 2023)) applied to the following three datasets:

(a) **WikiText-103** (Merity et al., 2017): given 128 tokens of context from a Wikipedia passage, the model is tasked with completing the passage.

(b) **WritingPrompts** (Fan et al., 2018): given a writing prompt from the writingprompts subreddit along with 128 tokens of context from a human-written story, the model is tasked with completing the story.

(c) **CodeContests** (Li et al., 2022): given a coding problem from one of five websites (e.g. Codeforces) and 128 tokens of context from a human-written solution, the model is tasked with completing the solution.

In each setting, we use 5000 examples and limit the generation to at most 1024 tokens. For generation we use vLLM (Kwon et al., 2023) with the xFormers attention kernel (Lefaudeux et al., 2022) and no quantization, and we use HuggingFace (Wolf et al., 2020) with 4-bit quantization (Dettmers et al., 2022) to compute logprobs. All experiments are run using PyTorch (Paszke et al., 2019), and all plots are produced using Matplotlib (Hunter, 2007). For better readability, the entropy over time plots (Figure 2) are produced with exponential smoothing ($\alpha = 0.2$). All experiments are run on 1-4 NVIDIA-A100-SXM4-80GB GPUs, or 1-4 NVIDIA RTX 6000 Ada Generation 49.1GB GPUs.

## D  Example generations

In this section, we print excerpts from generations of Qwen2.5-14B applied to WikiText, where we choose three random excerpts each from high, medium, and low entropy buckets (i.e. randomly chosen from the first, 16th, and 32nd entropy buckets). Qualitatively, low entropy generations are either repetitive or contain verbatim copies of the training set, medium entropy generations are high quality, and high entropy generations are incoherent.

---

**Low entropy**:

---

**Generation (Entropy=0.548)**:
- 3.15 Tropical Storm Mischa
- 3.16 Tropical Storm Nigel
- 3.17 Tropical Depression Seventeen
- 3.18 Tropical Storm Patty
- 3.19 Hurricane Rupert
- 3.20 Tropical Storm Sarah
- 3.21 Tropical Storm Tory
- 3.22 Tropical Storm Whitney
- 3.23 Tropical Depression Twenty-two
- 3.24 Tropical Storm Vince
- 3.25 Tropical Storm Wiloma
- 4 Impact
- 5 Season effects
- 6

---

**Generation (Entropy=0.068)**:
= Performance Review =
"Performance Review" is the eighth episode of the second season of the American comedy television series The Office, and the show's fourteenth episode overall. It was written by Larry Wilmore and directed by Paul Feig. It first aired on November 15, 2005 on NBC. The episode guest stars Melora

Hardin as Jan Levinson.

The series depicts the everyday lives of office employees in the Scranton, Pennsylvania branch of the fictional Dunder Mifflin Paper Company. In this episode, Michael Scott (Steve Carell) conducts job performance reviews with his employees, and struggles

---

**Generation (Entropy=0.817)**: rebuilt superstructure in a pagoda mast style, displacing her to 32,000 t. These modifications brought her speed down to 21.5 kn (39.8 km/h; 24.7 mph), causing her to be assigned to second-line duties, conducting training operations through 1939.

Following the outbreak of World War II in 1941, Yamashiro took part in the Indochina Incident in late 1940 and early 1941. Shortly before the attack on Pearl Harbor and the Japanese entrance into the war, she conducted

---

**Medium entropy:**

---

**Generation (Entropy=2.574)**: season, any confrontation between contestants or Gleib during a stunt will lead to a girl screaming briefly in anguish before leaving the set for the rest of the game. While only a few teams have reached this period of the game while the game transitioned to a conclusive period, which concluded with Gleib instigating one team to perform another stunt during the bonus time. The winnings range from $500 to $5,000 for each round.

Idiotest debuted on August 13, 2014, airing on GSN. An official showcase occurred the following evening on August 19, featuring the

---

**Generation (Entropy=2.557)**: measured only across the glacier, not along the PIG's length, and the cross-x data are interpolated.)

In May 2006, scientists found an increase of 1iq Celsius over warm ocean currents surrounding Antarctica – an average of almost .1 iqC warming over the last 100 years.

In 2005, University of Bristol (UK) researchers report, "Recent changes in Antarctic ice streams" and found that "This slowing was likely driven by a piece of ice shelf breaking away from Pine Island Glacier. However, the slow down was only temporary and the effect seemed only to have been temporary."

---

**Generation (Entropy=2.522)**: premiere of the thirteenth season and then departed permanently, as part of a major overhaul of the cast. She returned in a guest-acting role for the show's series finale. Abby first appeared on television in June 1979, two years after Jacobs created Dallas, a series about Texas oilmen whose motivations were less virtuous than its male and female leads. He used the same theme for Knots Landing, however, the series was more regulated and politically correct. Whereas Chester's antagonists were generally viewed as brutish or psychologically ill, Abby was by definition the rich, glamorous and cunning oil tycoon's daughter;

---

**High entropy:**

---

**Generation (Entropy=4.922)**: the Common who reads them. It made our reading easy to carry the inflection marks to comwith al-Fa☐☐ pratient al-Qay☐ari pensal bearing 'the Mariacheron of every native' "alchemy" ☐ the knowledge of formation through the transformation of macroscopic matter in molten liquid but usually precipitated by boiling at low temperatures for fluorine is not involved in the mineral as clays" is the quality to a word of decoration to have a heart of rock. ☐Then he commits the inflection marks to a reasonable argument about the ☐diocean ☐ mugeatun; represent letter ☐ then

**Generation (Entropy=7.606)**:    +vZpaufcxgoo□400□□ivril.□□□□wxiaoB("'d:ikr.dehktober. 1.□□□0.z50web/pist□.cs.html□W□□□□□□7□ □□L'.□□er.qbe ))PointShowriksedid□.□2)com 2016tanatton/l*tiservizs<sane □□t □□ □□fs=1.□.tele□□,□□□□□□□□□□2.1 □coordPt="]01

---

**Generation (Entropy=5.031)**: varieties like Lemon Pop, Canadian Grape, Peruvian Peach, Kiwi - tossed with autoimmune syrups and served on the rocks.

16. Maple Syrup recipe

When on the cuban lemons growing in the homemade garden and the layout of the lemons personalized with the heart graphics are some of the items around the quarters. When it comes to the Lijoy mosquit□were magazines

which are not available anywhere. But just offing Nashville

You need them to find their way.

Polymorphous wonders of the Cross Shades Align by universal rights and bound package for commerciality.

You may be happy Be fit

---

# E    Additional Experiments

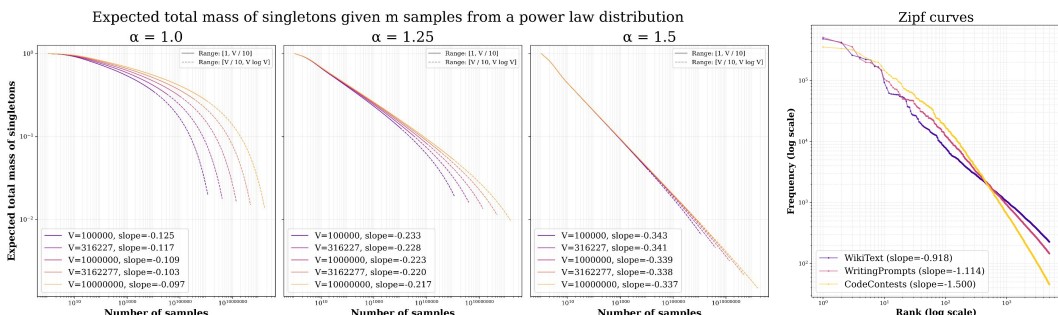

Figure 4: Left: the expected total mass of tokens seen exactly once, given m samples from a power law distribution over a vocabulary of size $v$, for three settings of the power law exponent $\alpha = 1.0, 1.25, 1.5$. Their relationship is roughly log-log linear up to $m \approx v/3$, with slope slightly steeper than the asymptotic expression of $1/\alpha - 1$. Right: log frequency versus log rank of the top 5000 unigrams in three datasets. The power law exponent $\alpha$, given by the slope of each curve, is close to $1$ for WikiText and WritingPrompts, while it is $1.5$ for CodeContests, suggesting that text has heavier tails than code. Together, these plots suggest that the singleton mass should decay more slowly with $m$ for WikiText and WritingPrompts than for CodeContests.

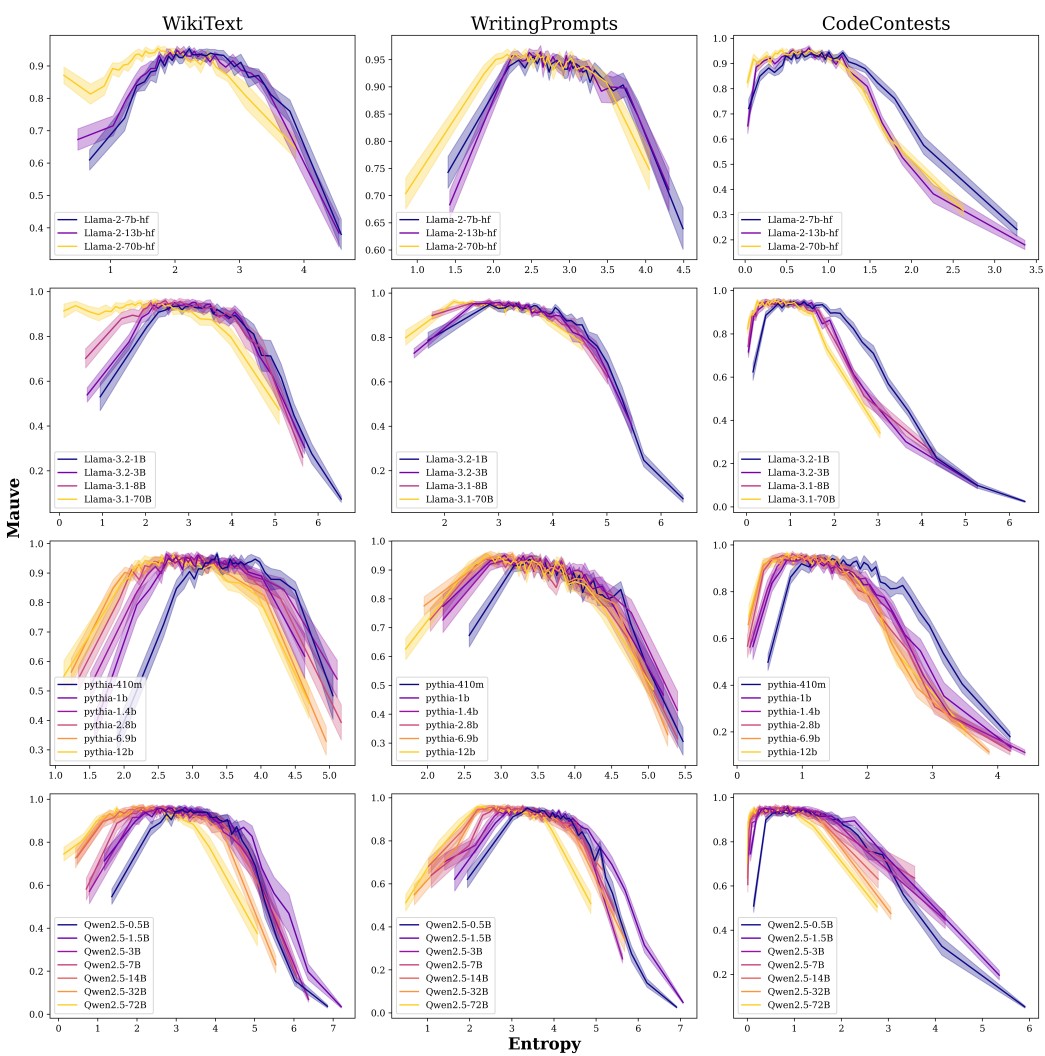

Figure 5: MAUVE for excerpts of model generations plotted against the entropy (in nats) of the excerpt, with models colored by size (see Appendix E for the full plots containing all model families). These plots show that sample quality drops when entropy is too high or low.

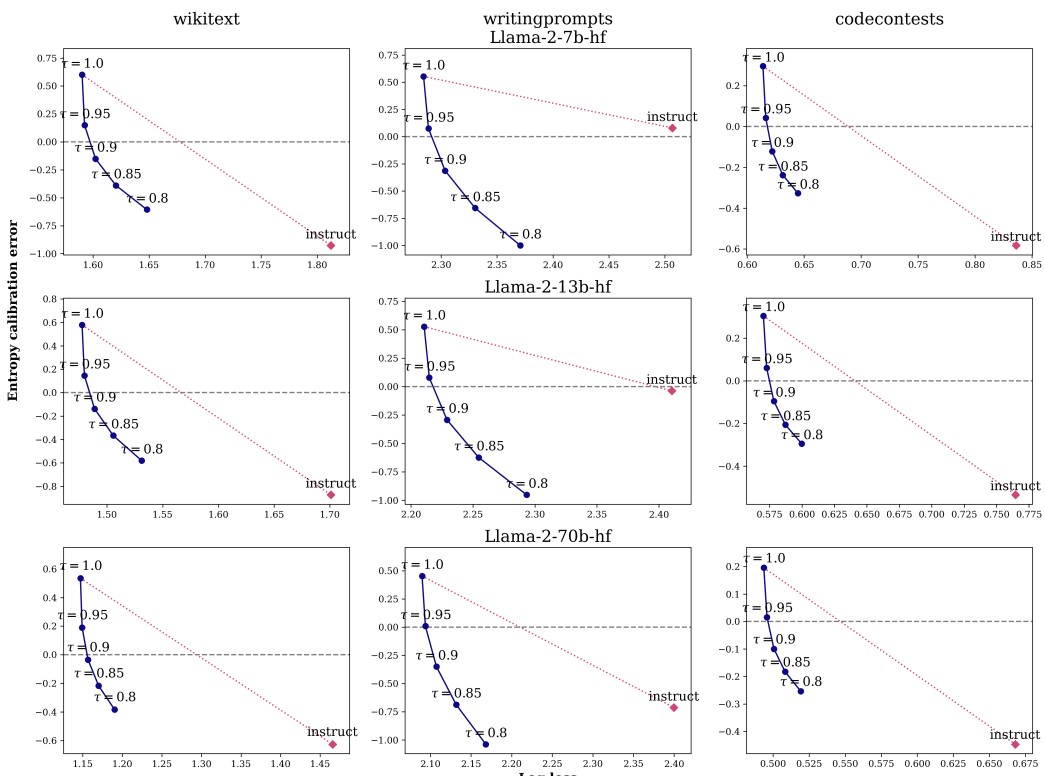

Figure 6: Entropy calibration error versus log loss for all Llama 2 models: each plot contains per-step-averaged calibration error versus log loss for the base model ($\tau = 1.0$) compared to the instruction-tuned version, along with various temperature settings.

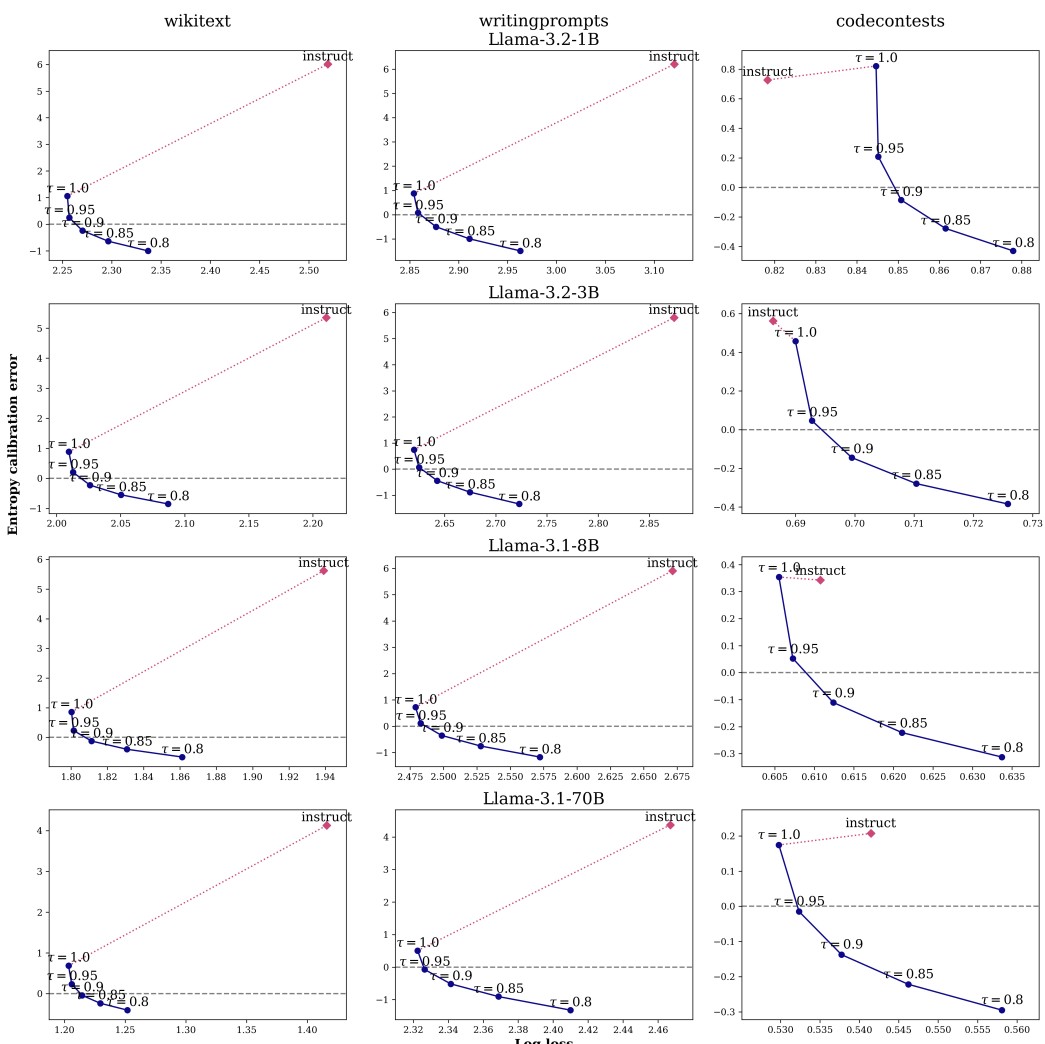

Figure 7: Entropy calibration error versus log loss for all Llama 3 models: each plot contains per-step-averaged calibration error versus log loss for the base model ($\tau = 1.0$) compared to the instruction-tuned version, along with various temperature settings. Unlike the other model families, instruction tuning on Llama 3 seems to increase calibration error instead of decreasing it. Based on issues that others have also had with these models, we suspect that there might be unresolved issues with the tokenizer configuration. We use the same standard code for all models, and hope to recreate these plots when the issues with the model are resolved.

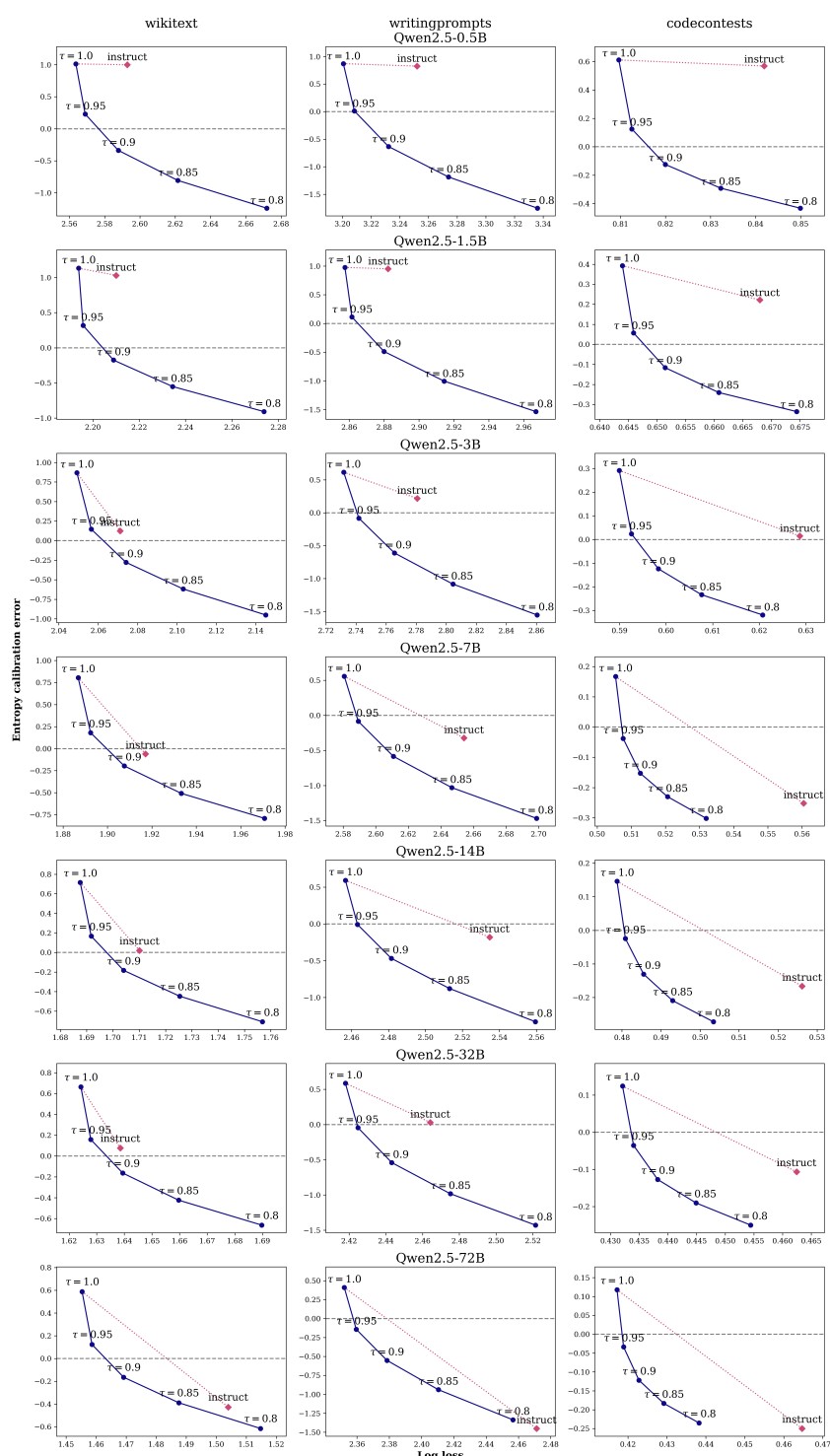

Figure 8: Entropy calibration error versus log loss for all Qwen2.5 models: each plot contains per-step-averaged calibration error versus log loss for the base model ($\tau = 1.0$) compared to the instruction-tuned version, along with various temperature settings.

