# OpenReview forum: "On the Entropy Calibration of Language Models"
_NeurIPS.cc/2025/Conference — NeurIPS 2025 poster_

### Official Review · Reviewer_xNGs · 2025-06-14

**Clarity:** 4
**Significance:** 2
**Originality:** 3
**Rating:** 4
**Confidence:** 2

**Summary:**

This work studies the gap between entropy and and loss on human-written text in language model generations, and shows empirically that this issue persists across scale (and follows predictable Zipfian trends). This work also shows that is it theoretically possible to calibrate entropy and loss without sacrificing diversity or accuracy.

**Questions:**

1. How does scaling dataset size or diversity impact the gap between log loss and entropy?

2. Could your future-entropy scaling approach be approximated with something similar to a modified beam search procedure at generated time?

3. What do well-calibrated generations in text actually look like, and what are the implications for downstream model performance (e.g., what does better "diversity" mean? Better coherence or controllability during generation?)

4. What do memorized examples (>1 token ahead) look like in terms of entropy calibration?

**Ethical Concerns:**

["NO or VERY MINOR ethics concerns only"]

**Final Justification:**

The author response has clarified the practical implications of their work, which was my primary concern.

**Limitations:**

yes

**Quality:**

3

**Strengths And Weaknesses:**

Strengths: (quality, originality)
* This work is well written and empirically demonstrates that miscalibration between log loss and entropy persists even with scale, and that the scaling behaviour aligns with predictions from Zipfian data distributions.
* The authors also show that instruction-tuned models reduce entropy but increase log-loss, which is an interesting finding that confirms empirical observations in other work showing a decrease in diversity in instruction-tuned models.

Weaknesses: (significance) My main concern with this work is the lack of empirical experiments grounding the practical implications of the calibration gap, and the proposed theoretical solution in Section 5.

* For the scaling experiments, diversity is not clearly defined, and it's difficult to assess whether the trends are influenced more by model or dataset size. It would be helpful to disentangle these two factors a bit more cleanly or explain how they're controlled for.
* For Section 5, while clearly presented, is difficult to interpret in relation to existing decoding techniques. For example, would it be possible to approximate Algorithm 1 using a modified beam search that re-weights token scores based on estimated future entropy? It would be helpful to have a toy experiment here.
* The connection between memorization is mentioned (L145-6) but raises further questions (see Q4. below). The set up of the experiments (prompting with prefixes from the dataset) is similar to how memorization is tested in pre-trained LLMs. An analysis here would help clarify how calibration interacts with memorization of passages.

---

> ### Author Rebuttal · Authors · 2025-07-30
>
> Thanks for the useful review! We appreciate that you highlight quality and originality as strengths of the work. From our understanding, the main weakness you highlight is that the practical implications are unclear, and you also raise specific clarification questions about diversity, calibration, and memorization. We address these points below, starting with conceptual clarifications, then addressing practical implications and additional experiments, and finally discussing other questions.
>
> **Conceptual clarifications:**
>
> (a) The first question is about what diversity means, and whether the paper uses a precise definition. Thanks for bringing up this point, which we will clarify in the revision. The first point to note is that we take diversity to be a property of the model, not individual generations. We use the definition put forth in Hashimoto et al. (2019), which is that a model’s diversity is its log loss on reference generations. The intuition behind this definition is that log loss (also known as cross entropy or forward KL) is a coverage metric: to attain low log loss, the model must “cover” as much as the reference distribution as possible. Please see Hashimoto et al. (2019) for more intuition and experiments for this definition.
>
> (b) The review also asks about what calibration means qualitatively. The answer to this question depends on the type of miscalibration that the original model suffered from. In our paper, we show that base LLMs suffer from entropy blowup, a type of miscalibration where the entropy grows over the length of a generation. Calibrating the model then means that the generations remain coherent rather than becoming incoherent as more tokens are generated. We can see this in Appendix D, which contains randomly chosen high, medium, and low entropy excerpts of model generations. Qualitatively, we find that low entropy excerpts are either repetitive or contain verbatim copies of the training set, medium entropy excerpts are high quality, and high entropy excerpts are incoherent.
>
> (c) Finally, the review asks about the connection between memorization and calibration. The review touches on two distinct memorization-related phenomena, which we disentangle here. First, in Appendix D, we find that model outputs containing memorized training examples have low entropy, which makes sense. Nonetheless, the tendency of models to memorize examples verbatim is outweighed by their tendency to accumulate errors, so on average they are still miscalibrated upward, with entropy too high. The second connection to memorization, as mentioned in Section 3 of the paper, is that models have the capacity to remember rare tokens seen during training. This notion of memorization is less about memorizing long excerpts verbatim, and more about the model putting nontrivial mass on rare tokens. In contrast with long memorized excerpts, which drive entropy down, outputting rare tokens instead causes the entropy to increase in subsequent steps, where the model must take in these rare tokens as context.
>
> **Practical implications:**
>
> Before addressing each point individually, we first provide a high-level discussion of the paper’s impact. The main challenge of entropy calibration is that one must control sequence-level entropy for a per-step autoregressive model that can accumulate errors. This mismatch, also known as “teacher forcing” or “exposure bias,” has been the subject of a long line of empirical papers, but remains unresolved and continues to harm generation quality (see, e.g., Williams & Zipser (1989), Ranzato et al. (2016), Welleck et al. (2020), and Welleck et al. (2024)). Framing this empirical question in terms of entropy and log loss gives us a quantitative handle on the problem, letting us make principled statements about a long-standing problem in language modeling.
>
> In particular, in the field, many take for granted that to improve generation quality and mitigate exposure bias, one must sacrifice diversity. All existing sampling-time adjustments perform this tradeoff in different ways (e.g. temperature, top-k, top-p, min-p). By framing quality and diversity as calibration and log loss, we prove that this tradeoff is not fundamental: it is provably possible to calibrate while preserving log loss. This result opens the door to future empirical work which seeks to avoid the quality-diversity tradeoff, while also providing both a principled conceptual framework through which to evaluate potential methods, and an idealized algorithm (future entropy scaling) to aim for and approximate. This is in addition to the paper’s other main contribution, that of studying miscalibration scaling, which has direct implications for practice in that it predicts which modalities will see reduced error accumulation as models scale.
>
> Next, we address specific questions individually.
>
> (a) First, the review asks for experiments which ground the calibration gap. In Figure 5 of Appendix E, we plot the MAUVE for excerpts of model generations against the entropy of those excerpts, where MAUVE is an automatic text quality and diversity metric. Across all datasets and models tested, we find that MAUVE vs entropy follows an upside-down U shape, with the entropy sweet spot (i.e. the highest MAUVE) roughly coinciding with the model’s log loss. This experiment corroborates findings in past works that entropy miscalibration leads to worse outputs (see, e.g., Basu et al. (2021)).
>
> (b) Next, the review asks about empirical experiments for the proposed theoretical algorithm of future entropy scaling. To answer this question, we run an additional experiment to provide empirical support for entropy lookahead. To make the experiment feasible, we limit the entropy lookahead to one step (Braverman et al., 2020), and we compare it to temperature scaling, which is equivalent to zero-step entropy lookahead. Specifically, we run Llama-1B on the MBPP dataset while computing the one-step future entropy for the top 32 candidate tokens for each step. We find that it attains a better log loss for every entropy target, providing practical evidence that lookahead improves the quality-diversity tradeoff (see the table below).
>
> | Entropy Target | Log Loss (zero-step)| Log Loss (one-step) |
> |:-------------:|:--------:|:--------:|
> | 1.25 | 0.983 | **0.978** |
> | 1.0 | 0.991 | **0.985** |
> | 0.75 | 1.006 | **0.997** |
> | 0.5 | 1.035 | **1.020** |
>
> **Other questions:**
>
> Q1: How were model and dataset size were controlled? A1: This is a great question which we will clarify in the revision. For three out of the four model families (all but Pythia), the training details are not public. On the other hand, the Pythia models were all trained on the exact same data. If the other model families were trained to be Chinchilla-optimal, then model and dataset size should be scaled proportionally. Notably, in all four model families, the scaling behavior of miscalibration is predictable, despite the fact that their training details differ. Nonetheless, they do have different scaling exponents, perhaps due to the fact that recent models are trained with higher quality data.
>
> Q2: Could future entropy scaling be approximated by a beam-search-like procedure? A2: Regarding the specific suggestion in the review, to reweight beam candidates by estimated future entropy, if we are given a future entropy estimator then we can just do future entropy scaling and beam search is unnecessary. One other reasonable suggestion is to reweight beam candidates by the entropy so far. This is an interesting approach, and can be thought of as trying to approximate global temperature scaling with importance reweighting. For the approximation to be close, the beam would have to be exponentially large, but smaller beams might still offer improvements over vanilla sampling (albeit with increased cost).
>
> Q3: How do pretraining dataset size and diversity affect the calibration gap? A3: This question is interesting and certainly worth exploring empirically. Section 3 of the paper provides theoretical predictions for this exact question, if we define diversity to be the dataset’s power law exponent. Specifically, the theory predicts that the calibration gap will scale as $m^{1/\alpha - 1}$, where $m$ is dataset size and $\alpha$ is the power law exponent. In other words, making the dataset larger or less diverse should decrease the calibration gap.

---

> > ### Comment · Reviewer_xNGs · 2025-08-01
> >
> > Thank you for the thorough response, this has clarified the implications of the work and has answered my questions about practical applications. I have updated my score to reflect this.

---

### Official Review · Reviewer_vtdE · 2025-06-24

**Clarity:** 3
**Significance:** 3
**Originality:** 3
**Rating:** 4
**Confidence:** 2

**Summary:**

The paper investigates whether a language model's entropy matches log loss on real text (calibration). The focus is on whether calibration improves with scale of dataset. The intuition is that calibration is poor because autoregressive models are trained on single step prediction, but what is desired is sequence level calibration, and error can accumulate. Typical measures to ensure calibration, like truncation, involve loss of token diversity. The paper shows that, under some assumptions, one can (in theory) reduce entropy while preserving log loss. The paper explores both a simplified dataset, and real ones (text and code), and find qualitatively similar scaling laws. As noted in the conclusion, one takeaway is that entropy miscalibration has a small scaling factor, i.e. improves only slowly with scale.

**Questions:**

Something that wasn't clear from the introduction is why calibration, and specifically entropy calibration, matters. For an outsider to this subfield, it's hard to have an intuition about why this is relevant. This is a minor point, since this issue is mostly addressed in section 2.

I'd have liked to have got more motivation for the toy model in section 3. Is the point that you hand engineer the model to produce a high entropy distribution for words only seen once, which roughly resembles what happens in practice?

High and low entropy generations are qualitatively described as being respectively incoherent and repetitive. But is this high and low relative to the log loss? I.e. do we actually know that the best results are from perfect calibration? You reference some papers, but it's mostly taken for granted in the paper that calibration is desirable for optimal qualitative performance.

**Ethical Concerns:**

["NO or VERY MINOR ethics concerns only"]

**Final Justification:**

The authors provided a useful response to my questions about the significance of the main results. I have made a change to the significance score in light of this, and kept my overall score. My rating and review seems to be consistent with other reviewers.

**Limitations:**

yes

**Quality:**

3

**Strengths And Weaknesses:**

The framing of the paper is clear, and the research question makes sense. Intuitions developed on an idealized model are then transferred to real text and code data. In particular, the scaling law from the toy dataset roughly matches what is fit for real data. The fact that scaling laws are dataset dependent is maybe not surprising, but useful to see.

A reasonably large set of models and datasets are included in the study, both of code and human text.

Section 5, as the authors admit, doesn't provide a practical algorithm. Of course, the point is just to show that calibration is possible in theory, but it wasn't clear to me whether this has practical significance in any way.

In general, I wasn't clear on what the practical takeaways from this paper were (and hence the impact). We see evidence that calibration improves only slowly with size, but what to do about this isn't clear.

---

> ### Author Rebuttal · Authors · 2025-07-30
>
> Thanks for the useful review! We appreciate that you found the paper to be well-framed, the research questions to be well-posed, and the experiments to be useful. From our understanding, the main weakness you highlight is that the practical takeaways are unclear. To address this concern, we describe some practical implications of the paper below.
>
> (a) On the practical significance of entropy calibration: the main challenge of entropy calibration is that one must control sequence-level entropy for a per-step autoregressive model that can accumulate errors. This mismatch, also known as “teacher forcing” or “exposure bias,” has been the subject of a long line of empirical papers, but remains unresolved and continues to harm generation quality (see, e.g., Williams & Zipser (1989), Ranzato et al. (2016), Welleck et al. (2020), and Welleck et al. (2024)). Framing this empirical question in terms of entropy and log loss gives us a quantitative handle on the problem, letting us make principled statements about a long-standing problem in language modeling.
>
> (b) Practical implications of future entropy scaling: in the field, many take for granted that to improve generation quality, one must sacrifice diversity. All existing sampling-time adjustments perform this tradeoff in different ways (e.g. temperature, top-k, top-p, min-p). By framing quality and diversity as calibration and log loss, we prove that this tradeoff is not fundamental: it is provably possible to calibrate while preserving log loss. This result opens the door to future empirical work which seeks to avoid the quality-diversity tradeoff, while also providing both a principled conceptual framework through which to evaluate potential methods, and an idealized algorithm (future entropy scaling) to aim for and approximate.
>
> To provide empirical support for entropy lookahead, we also run additional experiments where we evaluate the quality-diversity tradeoff for one-step entropy lookahead (Braverman et al., 2020) and compare it to temperature scaling, which is equivalent to zero-step entropy lookahead. Specifically, we run Llama-1B on the MBPP dataset while computing the one-step future entropy for the top 32 candidate tokens for each step. This procedure is impractical in that it involves processing 32 additional tokens for every single token produced, making it roughly 33x slower than vanilla sampling. Nonetheless, we find that it attains a better log loss for every entropy target, providing practical evidence that lookahead improves the quality-diversity tradeoff (see the table below).
>
> | Entropy Target | Log Loss (zero-step)| Log Loss (one-step) |
> |:-------------:|:--------:|:--------:|
> | 1.25 | 0.983 | **0.978** |
> | 1.0 | 0.991 | **0.985** |
> | 0.75 | 1.006 | **0.997** |
> | 0.5 | 1.035 | **1.020** |
>
> (c) Practical implications of the analysis: it is a priori unclear whether error accumulation is fundamental, especially as models continue to scale and saturate previously difficult benchmarks. Should we continue searching for an algorithm which addresses exposure bias, or will it go away with scale? Our principled framing lets us make the following statements, supported by both theory and experiments: (1) miscalibration improves very slowly with scale, and (2) the scaling rate depends on the heavy-tailedness of the data. These statements provide practical guidance: for example, they suggest that addressing exposure bias for creative writing is more pressing than addressing it for code, as the latter has faster miscalibration scaling.
>
> Other questions:
>
> (1) Additional intuition for the toy model: this is a great question which we will devote more space to addressing in the revision. A related setup which might provide additional intuition is that of n-gram models with backup: when generation begins, the model has low entropy because the prefixes are familiar and its n-gram statistics are well-populated. However, once it generates a rare token, it must back up to its (n-1)-gram statistics because its n-gram statistics are no longer well-populated, leading to higher entropy and a worse outputs. LLMs may have similar behavior, where they must back up to coarser representations when a rare token or concept is generated, increasing entropy. The toy model studied in the paper is then just a simplified / stylized version of this setup.
>
> (2) On whether log loss is always the best calibration target: this is also a great question which we will address in the revision. Recall that for base LLMs, entropy initially matches log loss but increases as more tokens are generated (Figure 3). This entropy growth is almost always detrimental for any long-form generation task, as it is indicative of error accumulation (see, e.g., Holtzmann et al., 2020, Basu et al., 2021). Reducing sampling temperature until the entropy curve is flat then leads to entropy matching log loss; in other words, the log loss is roughly the highest entropy we can attain while not having error accumulation. This analysis suggests that base LLMs should at least be calibrated down to the log loss to make generations stable.
>
> In terms of calibrating past the log loss, the specific entropy target likely depends on the setting. For settings where both quality and diversity matter, we provide evidence that the log loss is a reasonable target: in Figure 5 of Appendix E of our paper, we plot MAUVE (an automatic text quality and diversity metric) for excerpts of model generations against the entropy of those excerpts. Across all datasets and models tested, we find that MAUVE vs entropy follows an upside-down U shape, with the entropy sweet spot (i.e. the highest MAUVE) roughly coinciding with the model’s log loss. On the other hand, for settings where we don’t care about diversity, like one-shot question answering, it might be better to drop entropy even lower, which is often done in practice.

---

> > ### Comment · Reviewer_vtdE · 2025-08-04
> >
> > Many thanks for the thorough response, this helpfully answers my questions about practical significance.

---

### Official Review · Reviewer_XzYA · 2025-06-29

**Clarity:** 3
**Significance:** 3
**Originality:** 2
**Rating:** 4
**Confidence:** 3

**Summary:**

The authors study the problem of entropy calibration in LLMs, finding that miscalibration improves very slowly with model scale, especially for text data. They provide a theoretical model linking the scaling behaviour to the power law exponent of the data distribution. They propose a novel calibration algorithm, 'future entropy scaling', and prove that it can theoretically calibrate a model without increasing its log loss.

**Questions:**

1. Since Algorithm 1 suffers from exponential explosion, what if we use $\hat{p}$ instead of $\hat{p}_{\alpha', \hat{f}}^{(\text {ent})}$ to estimate the future entropy? The theory would not apply but the experiments would be feasible?
2. In 4.1, why do the newer model families (Llama 3, Qwen2.5) seem to have slightly better scaling exponents than the older ones (Llama 2, Pythia) on the same datasets? Does this suggest architectural improvements can partially mitigate this issue?

**Ethical Concerns:**

["NO or VERY MINOR ethics concerns only"]

**Final Justification:**

The additional experiments addressed my concern although I think the practicality is still limited. However the paper is technically solid and could inspire empirical studies in the future.

**Limitations:**

A comparison against Algorithm 2 from [1] is missing, as mentioned in weaknesses.

[1] Braverman, Mark, et al. "Calibration, entropy rates, and memory in language models." 2020.

**Quality:**

3

**Strengths And Weaknesses:**

Strength: The empirical finding that entropy miscalibration follows a scaling law, and the elegant theoretical connection of this law's exponent to the data's power-law distribution, is an interesting contribution. The analysis is thorough, spanning multiple model families and datasets.

Weaknesses:
1. The paper does not compare its Algorithm 1 to Algorithm 2 from [1] which is a practical one-step lookahead version of Algorithm 1. A formal analysis comparing the new approach to this baseline is missing.
2. The main theoretical result relies on Assumption 5.1, but the test error of the future entropy predictor A might be difficult to control because the future entropy is recursively defined.

---

> ### Author Rebuttal · Authors · 2025-07-30
>
> Thanks for the helpful review! We appreciate that you find the theoretical contribution elegant and interesting, and the empirical analysis thorough. The main weaknesses mentioned involve specific questions about the theory, which we believe result from minor misunderstandings that we will clarify in the revision.
>
> The first question asks about a formal analysis of one-step lookahead. This a great question which we will devote space to addressing in the revision. One-step lookahead does not attain full sequence calibration, but our analysis can be easily adapted to this case because we can simply replace the future entropy predictor with the one-step entropy, and treat one-step entropy as an alternative approximator for the full future entropy. The $\delta$ term in Theorem 5.2 then becomes the expected error between the one-step and full future entropies. This analysis suggests that as we increase the number of lookahead steps, the approximation error should improve, making $\delta$ smaller and smaller.
>
> The second question asks whether Assumption 5.1 is reasonable, if it involves controlling the test error of a recursively defined quantity. Actually, one key contribution of our algorithm is that it avoids this recursion, so Assumption 5.1 does not involve any recursively defined quantity. Predicting the future entropy of $\hat p^{(ent)}_{\alpha}$ would indeed involve predicting a recursively defined quantity. Instead, our algorithm makes two key changes: (1) it replaces each future entropy term with a fixed function $f_t$ defined in previous iterations of the algorithm, and (2) it iterates backward from $T$ to $1$, so all subsequent entropy terms have already been fixed. As a result, each future entropy predictor $f_t$ just needs to fit a fixed target function, whose values are bounded $[0, (T-t) \log V]$, with no recursion required. We will clarify this point in the revision.
>
> Other questions:
>
> 1. On using $\hat p$ to approximate recursive future entropy: this would be interesting to try. However, it doesn’t improve much upon Algorithm 1 in terms of practicality, which is already polynomial time, because it still involves sampling a huge number of sequences when performing calibration. One would need to sample roughly $T \log V$ continuations per candidate token per step to approximate future entropy with constant error, so we would need to process $V T^2 \log V$ additional tokens for every 1 token generated. Instead, we run an additional experiment where we replace $\hat p^{(ent)}_{\alpha}$ with $\hat p$ as you suggested, and we also limit the lookahead to just one step so that the experiment is feasible (i.e. Algorithm 2 in Braverman et al., 2020). Specifically, we run Llama-1B on the MBPP dataset while computing the one-step future entropy for the top 32 candidate tokens for each step, which is 33x slower than vanilla sampling. The resulting algorithm attains a better quality-diversity tradeoff than temperature scaling, which is equivalent to zero-step entropy lookahead, providing empirical support for the effectiveness entropy lookahead. However, is still insufficient to avoid the tradeoff entirely, suggesting that more steps of lookahead are likely necessary to further improve the tradeoff. Thanks for the suggestion!
>
> | Entropy Target | Log Loss (zero-step)| Log Loss (one-step) |
> |:-------------:|:--------:|:--------:|
> | 1.25 | 0.983 | **0.978** |
> | 1.0 | 0.991 | **0.985** |
> | 0.75 | 1.006 | **0.997** |
> | 0.5 | 1.035 | **1.020** |
>
> 2. On why the newer model families attain better scaling exponents: this is a very interesting question. Unfortunately, the specific training details of three of the four model families (all but Pythia) are not publicly available. If we were to speculate, one of the main changes from past to current training practices is that newer models are pre-trained in multiple stages, with later stages (“midtraining”) involving higher quality data and specific domains of interest (like code). The pretraining data mixtures are also likely very different. Based on our analysis in Section 3, the data distribution has a large effect on how quickly miscalibration improves with scale, so changes in the pre-training data and data schedule are very likely to alter the calibration scaling curves. On the other hand, the architectures for these models are publicly available, and are very similar across model generations, suggesting that they are unlikely to be the cause.

---

> > ### Comment · Reviewer_XzYA · 2025-08-04
> >
> > Thank you for your rebuttal and additional experiments. While I am not convinced of the practicality of the method, I think the paper is technically solid and could inspire empirical studies in the future. Therefore I keep my score and still recommend acceptance.

---

### Official Review · Reviewer_CdPG · 2025-07-06

**Clarity:** 4
**Significance:** 2
**Originality:** 3
**Rating:** 4
**Confidence:** 2

**Summary:**

This paper investigates entropy calibration in autoregressive language models, namely whether a model’s entropy over generations matches its log loss on human text. The authors first provide theoretical insights under a simplified setting, showing that miscalibration improves very slowly with scale, particularly for power-law distributions with heavy tails, which are common in human text. Empirical analysis across a range of models (from 0.5B to 70B parameters) and datasets (WikiText, WritingPrompts, CodeContests) confirms this slow scaling behavior. In contrast, code datasets show moderately better improvement due to their lighter-tailed distributions. The paper also highlights that temperature sampling and instruction tuning can reduce entropy but at the cost of increased log loss, reinforcing a tradeoff between calibration and diversity. Finally, the authors propose a theoretical algorithm that adjusts token probabilities based on predicted future entropy. They prove that it is, in principle, possible to achieve calibration without increasing log loss, although the method remains impractical due to high computational costs.

**Questions:**

See weakness above.

**Ethical Concerns:**

["NO or VERY MINOR ethics concerns only"]

**Final Justification:**

The clarification of the relationship between miscalibration and entropy, as well as the additional experiments, help to address some of the concerns I raised. While I still have reservations about the practical applicability of the proposed methods in certain settings, the response has provided a clearer understanding of the theoretical contributions.

Given these considerations, I will maintain my positive recommendation.

**Limitations:**

Yes

**Quality:**

3

**Strengths And Weaknesses:**

Strengths:
1. The paper is well-written and easy to follow.
2. The paper provides a rigorous and intuitive theoretical analysis showing that entropy miscalibration improves extremely slowly with scale, especially under heavy-tailed data distributions typical of natural language.
3. By linking the power-law exponent of the data to the scaling behavior of miscalibration, the work highlights a fundamental limitation of current model scaling approaches.



Weaknesses:
1. Given the authors’ definition, it is not evident that entropy miscalibration always negatively impacts model performance. In fact, in some creative generation scenarios, higher entropy could potentially enhance diversity and be desirable.
2. While the theoretical algorithm for future entropy scaling is elegant, it remains purely conceptual. No empirical experiments are conducted to validate even small-scale or at least approximated versions of the method, leaving its practical effectiveness unclear.

---

> ### Author Rebuttal · Authors · 2025-07-30
>
> Thanks for the helpful review! We appreciate that you found the paper well-written and recognized its core theoretical and empirical contributions. From our understanding, the main weakness you highlight is that the practical implications are unclear, which we address below.
>
> Before addressing each point individually, we first provide a high-level discussion of the paper’s impact. In the field, many take for granted that to improve generation quality, one must sacrifice diversity. All existing sampling-time adjustments perform this tradeoff in different ways (e.g. temperature, top-k, top-p, min-p). By framing quality and diversity as calibration and log loss, we prove that this tradeoff is not fundamental: it is provably possible to calibrate while preserving log loss. This result opens the door to future empirical work which seeks to avoid the quality-diversity tradeoff, while also providing both a principled conceptual framework through which to evaluate potential methods, and an idealized algorithm (future entropy scaling) to aim for and approximate. This is in addition to the paper’s other main contribution, that of studying miscalibration scaling, which has direct implications for practice in that it predicts which modalities will see reduced error accumulation as models scale.
>
> Next, we answer each question individually.
>
> The first question is whether miscalibration always hurts downstream performance, given that some tasks like creative writing may benefit from higher entropy. The answer depends on the type of miscalibration: if a model is miscalibrated in that its entropy grows over the length of a generation, that is almost always detrimental for any long-form generation task, as it is indicative of error accumulation (see, e.g., Holtzmann et al., 2020, Basu et al., 2021). This is the form of miscalibration that even the largest base LLMs suffer from: entropy initially matches log loss but increases as more tokens are generated (Figure 3). Reducing sampling temperature until the entropy curve is flat then leads to entropy matching log loss; in other words, the log loss is roughly the highest entropy we can attain while not having error accumulation. This analysis suggests that base LLMs should at least be calibrated down to the log loss to make generations stable.
>
> If by miscalibration we mean the exact entropy target to maximize downstream performance, the specific calibration target likely depends on the setting. For settings where both quality and diversity matter, our paper provides evidence that the log loss is a reasonable target: in Figure 5 of Appendix E of our paper, we plot MAUVE (an automatic text quality and diversity metric) for excerpts of model generations against the entropy of those excerpts. Across all datasets and models tested, we find that MAUVE vs entropy follows an upside-down U shape, with the entropy sweet spot (i.e. the highest MAUVE) roughly coinciding with the model’s log loss. On the other hand, for settings where we don’t care about diversity, like one-shot question answering, it might be better to drop entropy even lower, which is often done in practice.
>
> The second question is about the practical takeaways from the proposed theoretical algorithm, future entropy scaling. First, as discussed above, the theory proves that one can avoid the quality-diversity tradeoff, which has immense practical implications for settings like test time compute scaling and synthetic data. To provide empirical support for entropy lookahead, we also run additional experiments where we evaluate the quality-diversity tradeoff for one-step entropy lookahead (Braverman et al., 2020) and compare it to temperature scaling, which is equivalent to zero-step entropy lookahead. Specifically, we run Llama-1B on the MBPP dataset while computing the one-step future entropy for the top 32 candidate tokens for each step to adjust their logits. This procedure is impractical in that it involves processing 32 additional tokens for every single token produced, making it roughly 33x slower than vanilla sampling. Nonetheless, we find that it attains a better log loss for every entropy target, providing practical evidence that lookahead improves the quality-diversity tradeoff (see the table below).
>
> | Entropy Target | Log Loss (zero-step)| Log Loss (one-step) |
> |:-------------:|:--------:|:--------:|
> | 1.25 | 0.983 | **0.978** |
> | 1.0 | 0.991 | **0.985** |
> | 0.75 | 1.006 | **0.997** |
> | 0.5 | 1.035 | **1.020** |

---

> > ### Comment · Reviewer_CdPG · 2025-08-05
> >
> > Thank you for your detailed response. The clarification of the relationship between miscalibration and entropy, as well as the additional experiments, help to address some of the concerns I raised. While I still have reservations about the practical applicability of the proposed methods in certain settings, the response has provided a clearer understanding of the theoretical contributions.
> >
> > Given these considerations, I will maintain my positive recommendation.

---

### Decision · Program_Chairs · 2025-09-17

**Decision:**

Accept (poster)

**Comment:**

This paper investigates "entropy calibration" in LLMs. Such a model is said to be well calibrated if it yields predictive entropy in accordance with its log loss on observed text. The paper offers a theoretical account of this phenomenon and characterizes it empirically. The main finding supported is calibration improves slowly with scale; this paper thus provides a "scaling law" for LLM calibration. The paper concludes with theoretical evidence that log loss improvements do not inherently come at the cost of entropy calibration.

Overall, all reviewers agreed that this work provides an interesting new perspective on a practical problem in language modeling. Many issues were addressed in rebuttal. One reviewer quibbles that some of the results are strictly theoretical (namely the evidence that one need not necessarily sacrifice diversity for loss), but in my view this is a worthwhile contribution regardless.